# Zyxin regulates embryonic stem cell fate by modulating mechanical and biochemical signaling interface

Songjing Zhang [1], Lor Huai Chong[2,3], Jessie Yong Xing Woon [1], Theng Xuan Chua[1], Elsie Cheruba [4], Ai Kia Yip[2], Hoi-Yeung Li [1], Keng-Hwee Chiam [2✉] & Cheng-Gee Koh [1✉]

Biochemical signaling and mechano-transduction are both critical in regulating stem cell fate. How crosstalk between mechanical and biochemical cues influences embryonic development, however, is not extensively investigated. Using a comparative study of focal adhesion constituents between mouse embryonic stem cell (mESC) and their differentiated counterparts, we find while zyxin is lowly expressed in mESCs, its levels increase dramatically during early differentiation. Interestingly, overexpression of zyxin in mESCs suppresses Oct4 and Nanog. Using an integrative biochemical and biophysical approach, we demonstrate involvement of zyxin in regulating pluripotency through actin stress fibres and focal adhesions which are known to modulate cellular traction stress and facilitate substrate rigidity-sensing. YAP signaling is identified as an important biochemical effector of zyxin-induced mechanotransduction. These results provide insights into the role of zyxin in the integration of mechanical and biochemical cues for the regulation of embryonic stem cell fate.

[1] School of Biological Sciences, Nanyang Technological University, Singapore, Singapore. [2] Bioinformatics Institute A*STAR, Singapore, Singapore. [3] School of Pharmacy, Monash University Malaysia, Subang Jaya, Malaysia. [4] Mechanobiology Institute, Singapore, Singapore. ✉email: chiamkh@bii.a-star.edu.sg; cgkoh@ntu.edu.sg

Mouse embryonic stem cells (mESCs) are unspecialized cells derived from blastocyst's inner cell mass. They possess the capability of self-renewal and are able to differentiate into many cell types[1,2]. Extensive investigations on mESC differentiation have led to a better understanding of the developmental pathways and their potential applications in regenerative medicine[3,4]. Apart from the well-documented biochemical signaling pathways[5], biophysical cues from the microenvironment also play important roles in ESC maintenance and differentiation[6]. Amongst the biophysical cues, traction stress and substrate stiffness are found to profoundly influence stem cell status[7-10]. Integrin-focal adhesion mediated modulation of multipotent adult stem cell and mesenchymal stem cell specification is a well-known example of stem cell fate regulation by mechanotransduction pathways[11-13]. However, how these mechanical cues stimulate intracellular signaling pathways in ESCs, and how mechanical and biochemical signals work together to regulate stem cell fate are not well understood.

Integrin-based focal adhesion (FA) complexes are composed of multiple FA proteins that form mechanical links between the actin cytoskeleton and the extracellular matrix[14,15]. Much of our current insights into the roles of FAs in mechanobiology are from studies in specialized cells. In specialized cell types, the core FA region is stratified into three layers[16]: (1) integrin signaling layer represented by FAK and paxillin, (2) force-transduction layer represented by talin and vinculin, and (3) actin regulatory layer represented by zyxin, VASP, and α-actinin. The molecular architecture of the FA complex has recently been mapped in mESCs[17]. The differences in FA nanostructure between mESCs and specialized cells raise the possibility that FA-mediated mechanotransduction in mESCs is different. In particular, expressions of LIM-domain FA proteins in mESCs were found to be lower compared to that in specialized cell types.

Zyxin is a LIM-domain protein found to localize to the FA, adheren junction and actin stress fibers[18-20]. It has two main domains: (1) N-terminal proline-rich domain, which forms complexes with actin regulators, VASP, and α-actinin; and (2) C-terminal region containing three conserved LIM domains responsible for force sensing, facilitating its recruitment to FAs[21]. The presence of a nuclear export signal in zyxin suggests that it might participate in communication between FA and the cell nucleus[22,23]. In addition, zyxin has been reported to be a mechano-sensor[21]. The recruitment of zyxin to FAs and adheren junctions is force-dependent[24,25] and so is the shuttling of zyxin between the cytoplasm and the nucleus[26]. Despite zyxin's multiple roles in specialized cells, its role in ESC remains largely unknown.

In this study, we investigated the crosstalk between mechanical and biochemical cues mediated by FAs in regulating embryonic stem cell fate. To preclude mechanical influence from E-cadherin-mediated cell-cell interaction, mouse embryonic stem cells D3[27] and E14[28] were chosen as two independent model systems, because of their ability to survive as single cells without loss of pluripotency[17]. Consistent with previous reports, our studies revealed that actin and FAs in differentiated cells are more matured and dynamic as compared to those in pluripotent stem cells. We further demonstrate that overexpression of zyxin in mESCs leads to a reduction in pluripotent markers Oct4 and Nanog. Concurrently, high zyxin levels also strengthen the actin cytoskeleton and FAs. Our investigations into zyxin's role in ESC differentiation suggest that zyxin modulates cellular traction stresses and substrate rigidity sensing in ESCs, and also regulates mechanosensitive effectors such as YAP and its downstream targets, leading to alterations in the transcription of pluripotency-maintenance genes such as Oct4.

## Results

### Actin and focal adhesions showed increased abundance and growth during early differentiation of mESC.

We wished to study the differences in the actin cytoskeleton and FAs in pluripotent stem cells and cells undergoing early-stage differentiation. To set up the pluripotent mESCs (ESCP) and their differentiated counterpart (ESCD), mESCs were treated with all-trans retinoic acid (RA, 2–10 μM) to induce mESC differentiation[29]. To monitor RA-induced early differentiation, levels of Oct4 and Sox2 pluripotency markers were determined over a 72 h time course. As high concentrations of RA induce cell death[30], 5 μM of RA treatment over 72 h was chosen as the optimal condition for induction of mESC differentiation. Progressive decline in Oct4 and Sox2 levels was observed during RA treatment in both D3 (Supplementary Fig. 1a) and E14 (Supplementary Fig. 1b) cells. Mouse embryonic fibroblasts (MEF) were used as specialized cell type for comparison (Fig. 1a). Pluripotent D3 (Supplementary Fig. 1c) and E14 (Supplementary Fig. 1d) cells (ESCP) were maintained in medium supplemented with leukemia inhibitory factor (LIF) to suppress spontaneous differentiation. While pluripotent D3 and E14 cells formed compact and multicellular colonies characterized by distinct colony borders, early differentiated D3 (Supplementary Fig. 1c) and E14 (Supplementary Fig. 1d) under RA treatment (ESCD) formed collapsed colonies with multipolar and elongated cells at the colony borders.

To compare the actin and FA structures that provide mechanical interaction between the cell and its microenvironment, ESCP and ESCD were seeded as single cells on fibronectin-coated coverslips and immuno-stained to observe actin and vinculin (Fig. 1b, c). Both ESCP and ESCD were much smaller compared to MEF. While actin and FAs were prominent in MEF, actin staining was weaker and FAs were smaller in mESCs (Fig. 1b). In general, ESCD (+RA) was more elongated and larger in cell area compared to ESCP (+LIF) (Fig. 1b, d). Actin was primarily localized at the cell periphery in ESCP, whereas stronger actin fibers were localized at cell periphery of ESCD, along with the presence of thinner intracellular actin filaments (Fig. 1b). Integrated actin intensity was found to increase in ESCD (Fig. 1e). As shown in the zoomed-in pictures (Fig. 1c), FAs appeared as small puncta and primarily distributed along the cell periphery in ESCP. In contrast, ESCD displayed big and elongated FAs, which strongly co-localized with the ends of stress fibers at the cell periphery. The number and area of FA were found to be higher in ESCD (Fig. 1f, g). Altogether, our data suggest that there are increase in stress fiber amount and growth of FAs, when D3 and E14 cells undergo RA-induced early differentiation.

### Drastic increase in Zyxin is observed during early differentiation.

The observations above led us to question what could have contributed to the differences in the actin and FA structures of mESCs versus those in cells which enter differentiation. The FA life cycle encompasses initiation, maturation, and disassembly[31]. Different FA proteins are recruited in sequential manner throughout the FA life cycle[32]. Based on the core FA proteins found in mESCs[17], we proceeded to compare the composition of FAs across a panel of mESCs (D3 and E14), hESCs (H1 and hiPSC), mouse fibroblast (NIH/3T3) and human fibroblast (BJ-5ta). We found that the levels of zyxin were consistently much lower in mESCs and hESCs compared with their respective differentiated cell types (Fig. 2a). In particular, zyxin was found at very low levels in mESCs (Fig. 2a and Supplementary Fig. 2a). This is consistent with an earlier study which reported that LIM-domain containing FA proteins are lower in abundance in mESCs[17]. Real-time PCR was adopted to verify the mRNA expression of zyxin in mESC. Data from two independent sets of

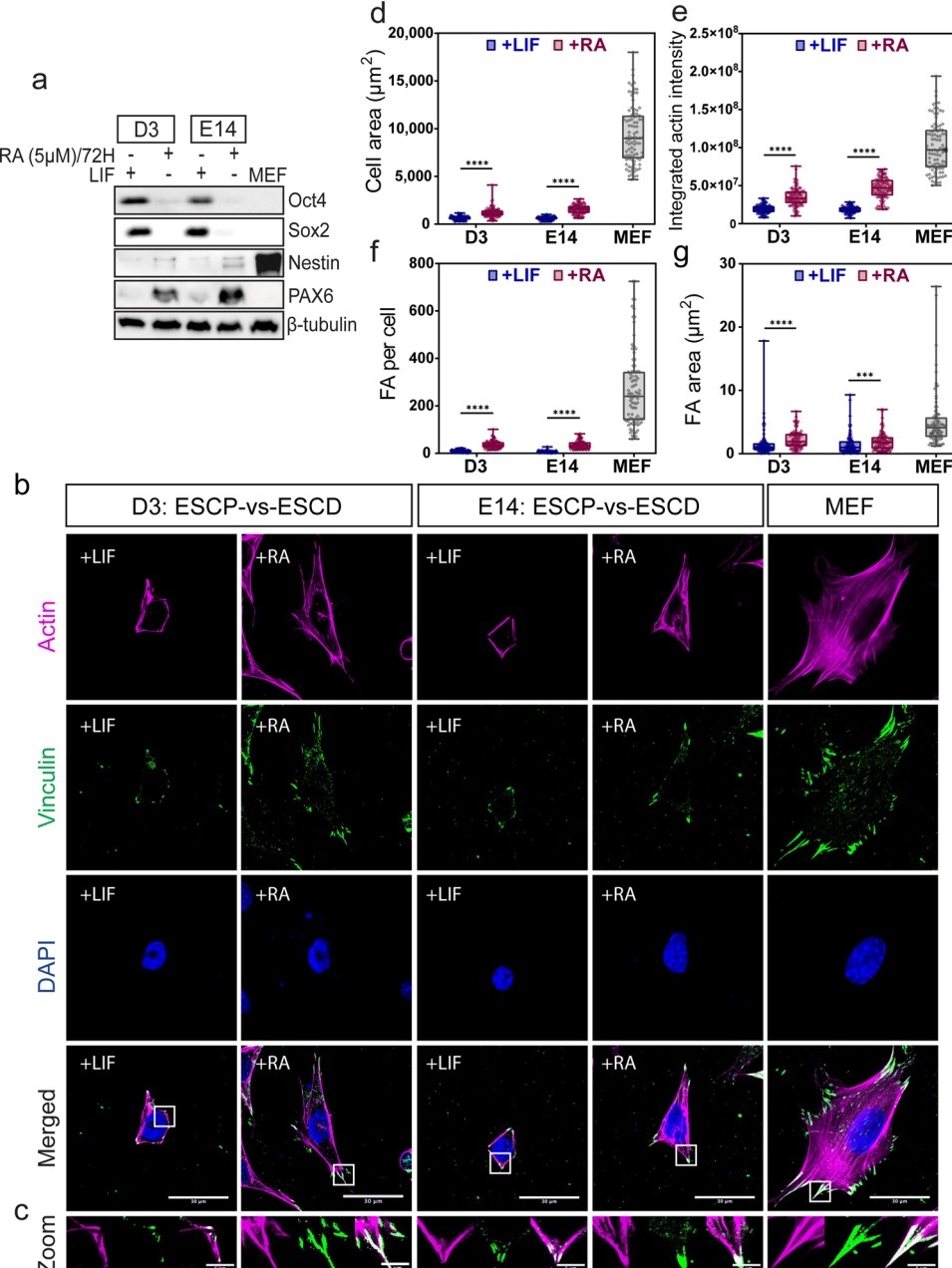

**Fig. 1 Comparison of focal adhesions and actin structures between pluripotent mESC (ESCP) and RA-induced differentiated mESC (ESCD). a** D3 and E14 cells were treated with retinoic acid (RA) for up to 72 h at a concentration of 5 μM. Leukemia Inhibitory factor (LIF) was used to maintain pluripotency of mESC. MEF was used as the specialized cell type for comparison. Pluripotency markers (Oct4, Sox2) and early differentiation markers (Nestin, Pax6) were examined by western blot. β-tubulin was used as loading control. **b** Examination of focal adhesion and actin cytoskeleton were conducted by immunofluorescence, using antibodies against vinculin (green) and Phalloidin-Alexa 546 to visualize focal adhesions and actin (magenta), respectively. **c** Boxed regions in merged picture (**b**) were zoomed in and shown here. The experiments were conducted in triplicates. Z-stacked images with maximum intensity projection were shown. Scale bar: 30 μm for original images, 5 μm for zoomed-in images. **d** projected cell area, **e** integrated actin intensity, **f** number of focal adhesions per cell, **g** projected focal adhesion area were measured ($N = 90$). Quantified data were averaged from three independent experiments. Two-tailed unpaired Student's $t$-test and Mann–Whitney $U$-test were used to test the differences between the +LIF and +RA groups. In the box and whisker plots, the center line is the median, the box-bounds are the 25th and 75th percentiles, and the whiskers are the 0.05 and 0.95 percentiles. ***$P \leq 0.001$; ****$P \leq 0.0001$.

zyxin primers consistently showed that zyxin transcript could be detected in mESCs but was ~1000-fold lower than that in embryonic fibroblasts (Fig. 2b). Due to its low abundance, we are thus not able to detect the presence of endogenous zyxin at the FA of mESCs, whereas clear zyxin localization at the FA was observed in fibroblasts (Fig. 2c, d). Xia et al. showed endogenous zyxin localization to FAs and concluded that mESCs are capable

of forming mature FAs[17]. However, we were not able to observe this in our imaging experiments which could suggest that the FAs formed in mESC are less mature compared to adherent cell types.

We went on to examine FA protein expression profiles during RA-induced early differentiation. D3 and E14 cells were treated with RA over a 72-hour time course, and changes in core FA proteins were analyzed by western blot. Zyxin levels increased

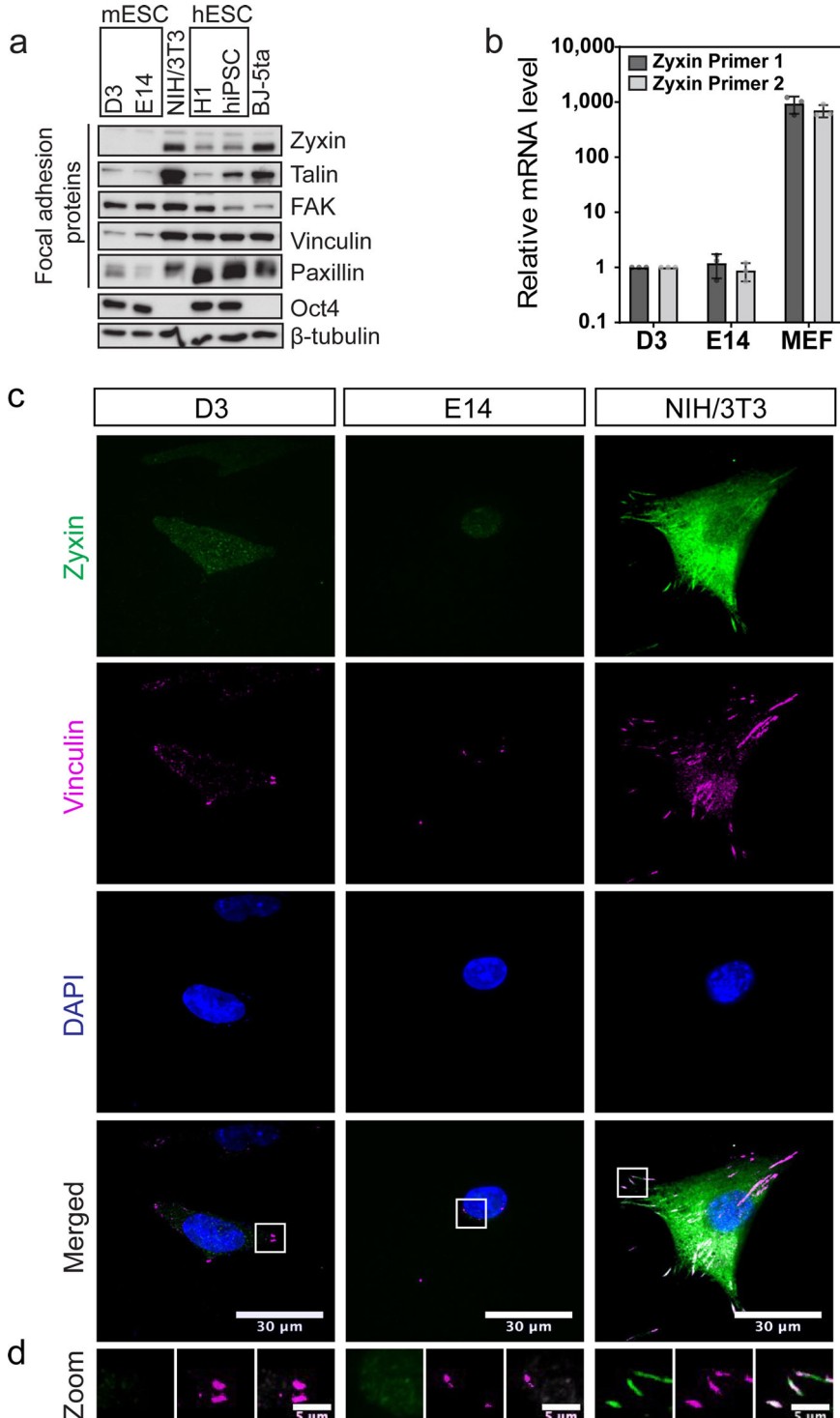

**Fig. 2 Zyxin is lowly expressed in both mESC and hESC. a** A panel of focal adhesion proteins were examined in mESC, hESC, mouse fibroblasts (NIH/3T3) and human fibroblasts (BJ-5ta). Oct4 was used to indicate pluripotency and β-tubulin was used as loading control. **b** Zyxin expression at mRNA level was monitored using two independent zyxin primer sets (N = 3). Relative quantification values were normalized against β-tubulin and presented as fold change with D3 as internal control. Error bars represent standard deviations. **c** Zyxin localization at focal adhesion was monitored by immunofluorescence, using antibody against zyxin. Vinculin was used as a marker for focal adhesion. **d** Focal adhesions at the cell periphery (boxed region in **c**) were zoomed in and shown here. Z-stacked images with maximum intensity projection were shown. Representative images from three biological repeats were shown. Scale bar: 30 μm for original images, 5 μm for zoomed in images.

drastically when D3 cells were treated with RA (Fig. 3a, b and Supplementary Fig. 2a, b). We also began to detect zyxin localization to the FA in RA-treated cells (Fig. 3c, d). Co-localization of zyxin and vinculin was observed at the FA of RA-

treated cells (Fig. 3e). Earlier studies have found that mechanical force-induced actin polymerization at FA promotes zyxin accumulation at the adhesion area[20,33]. Therefore, the increased zyxin accumulation at FA observed in early differentiated cells

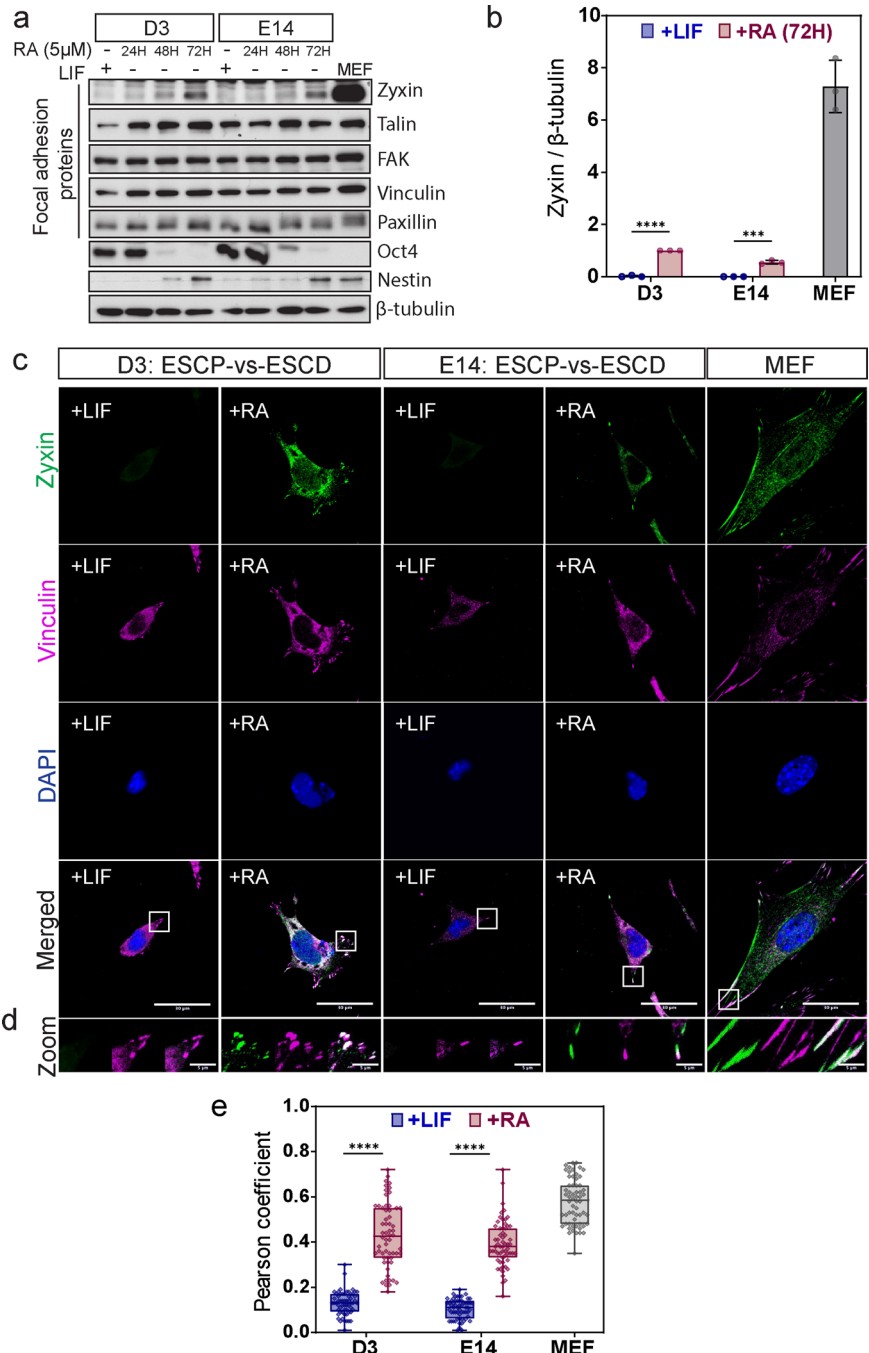

**Fig. 3 Zyxin level is elevated in the process of RA-induced early differentiation. a** D3 and E14 cells were treated with retinoic acid (RA) for 24 h, 48 h, 72 h, LIF was used to maintain pluripotency. A panel of focal adhesion proteins were examined. Pluripotency markers (Oct4) and differentiation markers (Nestin) were examined by western blot. β-tubulin was used as loading control. **b** Zyxin intensities from western blot were statistically quantified and plotted (N = 3). Values were normalized against β-tubulin. Results were averaged from three biological repeats. Two-tailed unpaired Student's t-test was used to test the differences between +LIF and +RA. Error bars represent standard deviations. **c** D3 and E14 were immunostained with zyxin antibody to examine zyxin localization. Vinculin was used as a marker for focal adhesion. Z-stacked images with maximum intensity projection were shown. Representative images from three biological repeats were shown. **d** Zoomed-in view of zyxin at focal adhesion was shown here. Scale bar: 30 μm for original images, 5 μm for zoomed in images. **e** Classical colocalization analysis of zyxin and vinculin was conducted from three independent experiments (N = 60). Pearson coefficient method was applied. Two-tailed unpaired Student's t-test and Mann–Whitney U-test were used respectively to test the differences between the +LIF and +RA groups. In the box and whisker plots, the center line is the median, the box-bounds are the 25th and 75th percentiles, and the whiskers are the 0.05 and 0.95 percentiles. ***P ≤ 0.001; ****P ≤ 0.0001.

could suggest possible roles for zyxin in mechanotransduction and regulation of stem cell fate.

**Zyxin negatively regulates pluripotency genes, and positively regulates actin and focal adhesion.** To further address the role of

zyxin in regulating stem cell fate, mCherry-Zyxin plasmid was introduced at increasing concentrations (0.1, 0.2, 0.5, 1.0, and 2.0 μg) in D3 and E14 (zyxin O/E), and cells transfected with 2.0 μg mCherry vector plasmid were used as control (ctrl). Transfected cells were checked for the expression of pluripotency

markers (Oct4 and Nanog). Overexpression of zyxin did not affect the levels of other FA proteins (Fig. 4a, b). However, we found gradual reduction of Oct4 and Nanog levels with increasing zyxin overexpression in D3 (Fig. 4a) and E14 (Fig. 4b), suggesting that zyxin negatively regulates the levels of pluripotency markers (Fig. 4c). To verify the effect of zyxin on pluripotency markers, mRNA levels of Oct4 and Nanog were also analyzed. In agreement with the western blot analysis, we observed reduced transcript levels of pluripotency markers in cells overexpressing zyxin (Fig. 4d). These results suggest that zyxin negatively regulates pluripotency markers in mESCs.

The observations that zyxin overexpression leads to reduction of pluripotent markers (Fig. 4a–d) prompted us to examine if zyxin knockdown could maintain pluripotency under RA-induced differentiation. Zyxin knockdown was achieved by transient transfection of siRNA with similar approach and knockdown efficiency reported in our previous work[34]. Consistently, we observed induction of zyxin expression in cells treated with RA but not in cells transfected with siRNA (Supplementary Fig. 3a–d). Taking reference from previous studies[35,36], we next proceeded to titrate RA treatment from 5 μM, 2 μM, to 1 μM in order to better observe the effects of zyxin knockdown. Oct4 levels were monitored over a 72-h time course in response to zyxin knockdown. At 72H time point, while RA-treatment led to significant reduction of Oct4 in control cells, Oct4 levels were better maintained in RA-treated zyxin-knockdown cells (Supplementary Fig. 3e). These observations suggest possible roles for zyxin in pluripotency maintenance.

Much of our current insights on the roles of zyxin in the regulation of actin and FAs are from studies performed in specialized cells[24,37]. Little is known about the roles of zyxin in mESCs. We next examine the effects of zyxin overexpression on actin and FA in pluripotent cells. Pluripotent D3 and E14 maintained in medium supplemented with LIF were transiently transfected with mCherry-Zyxin. To preclude mechanical influence from E-cadherin-mediated cell-cell interaction, transfected cells were plated on fibronectin-coated coverslips for 6 hours to allow for single cell imaging. Cells were stained with Alexa 488-phalloidin to visualize actin, and immuno-stained with vinculin to visualize the FA (Fig. 5a–d). D3 and E14 cells, which overexpressed zyxin, were more well spread and appeared more polarized as compared to control cells. Dense actin fibers were also observed at the cell periphery of zyxin-transfected cells (Fig. 5a, b, e). The integrated actin intensity was significantly increased in cells with zyxin overexpression (Fig. 5f). mCherry-Zyxin was found at small puncta, which co-localized with vinculin (Fig. 5c, d). The number and area of FAs were also significantly increased by zyxin overexpression (Fig. 5g, h). Our observations were consistent in two independent mESC cell lines, D3 and E14. To rule out the possibility of matrix specific effects, the above experiments were repeated using a different matrix, laminin. We obtained similar results when cells were plated on laminin (Supplementary Fig. 4). mCherry-Zyxin consistently localized to FAs in the overexpression cells (Supplementary Fig. 4a–d). Zyxin over-expressing D3 and E14 cells were more well spread and exhibited increased actin staining as well as FA numbers and areas (Supplementary Fig. 4e–h). Together, we found that high zyxin level downregulates pluripotency markers (Fig. 4) and concurrently increases actin and FAs (Fig. 5 and Supplementary Fig. 4) in pluripotent cells.

**Zyxin overexpression alters traction stress and facilitates substrate rigidity sensing**. Next, we determine if zyxin could regulate embryonic stem cell fate. In specialized cells, zyxin is recruited last to the adhesion sites to promote FA maturation and to facilitate internal force transmission and external force sensing[24]. Since traction stress and substrate sensing are important regulators of early fate decisions in mESCs[7,9], we hypothesize that zyxin may function as a mechanosensor to regulate stem cell fate through the modulation of actin cytoskeleton-mediated traction stress and substrate rigidity sensing in mESCs.

Before we establish the role of zyxin in the regulation of traction stresses, we first compared the basal 3D traction stress profile[38] (both lateral and axial stresses) exerted by cells plated on a 2D planar surface of three cell types: pluripotent cells (ESCP), early differentiated cells (ESCD), and specialized cells (MEF). Fibronectin-coated polyacrylamide (PAA) gels of 14 kPa were used as in our previous work[34]. Traction stresses were measured and calculated based on previously documented protocols[39]. MEF cell type exhibits high average in-plane traction stress magnitude (Txy) and low average out-plane stress magnitude (Tzz), whereas mESC cell type exhibit low Txy and high Tzz. RA-induced differentiation led to changes in Txy and Tzz of mESC (Supplementary Fig. 5). The average Txy exerted by the D3 (+RA) was about 1.5 times higher than that exerted by D3 (+LIF) (Supplementary Fig. 5a, b). The average Tzz exerted by the D3 (+RA) was about 2 times lower than that exerted by D3 (+LIF) (Supplementary Fig. 5c). The difference in the overall traction stress profiles between fibroblasts and pluripotent cells is due to the presence of stress fibers that terminate at the FAs which in turn generate substantial in-plane traction forces[40]. Accordingly, the increase in stress fibers and FAs (Fig. 1b) in ESCD (+RA) resulted in traction stress magnitude profile closer to that observed in fibroblasts.

To test our hypothesis, we examine the effects of zyxin overexpression on the mESC traction stress profile. Cells were transfected with mCherry-Zyxin and subjected to traction stress analysis. The magnitudes of the Txy exerted by zyxin over-expressing D3 and E14 cells were about 1.5 times higher than that exerted by control cells (Fig. 6a–c) while the magnitudes of Tzz were about 2 times lower than control cells (Fig. 6a, b, d), indicating traction stress profile of zyxin overexpressing cells is similar to that of early differentiated cells (Supplementary Fig. 5). These results suggest that mCherry-Zyxin localization to the FA (Fig. 5c, d) likely induces FA maturation and stress fiber formation (Fig. 5a, b) which in turn promote FA dynamics leading to an increase in in-plane (Txy) normal traction stress magnitudes (Fig. 6c) as well as a decrease in out-plane (Tzz) rotational traction stress magnitudes (Fig. 6d).

In response to elevation in substrate rigidities, specialized cells exert larger traction stresses associated with an increase in cell spreading on stiffer substrates[41,42]. However, mESC does not change apical cell stiffness and spreading in response to stiffer substrates. Since the FA and actin stress fiber structures are less well defined in mESC, it is possible that rigidity sensing is not as well developed in mESC due to the lack of actin bundles[43]. To investigate if zyxin could influence mESC substrate rigidity sensing, we analyzed stress magnitudes exerted by control and zyxin-overexpressing cells plated on fibronectin-coated gels with increasing substrate rigidity (6, 14, 31 kPa). The range of rigidities chosen was based on our previous work[34]. D3 control cells plated on increasing substrate rigidity displayed no significant changes in traction stress magnitudes at both in-plane (Txy) (Fig. 6e, f) and out-plane (Tzz) directions (Fig. 6g, h). However, in response to elevated substrate rigidities, zyxin-overexpressing D3 cells were able to sense substrate rigidities by increasing Txy magnitudes (Fig. 6e, f) and decreasing Tzz magnitudes (Fig. 6g, h). The results were reproducible in E14 control-vs-zyxin overexpression cell lines (Supplementary Fig. 6a–d).

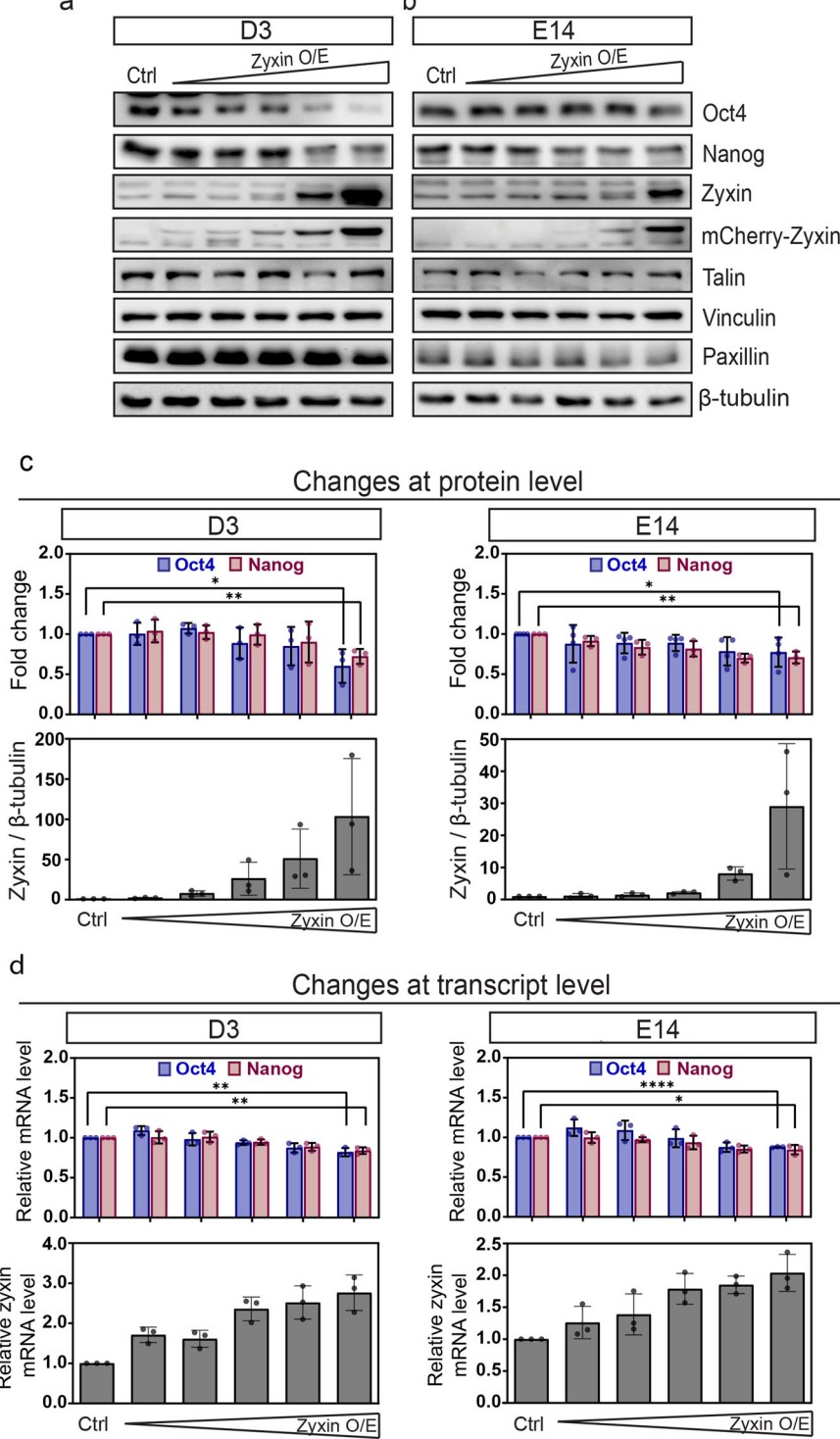

**Fig. 4 Zyxin overexpression supresses pluripotency makers. a**, **b** mCherry vector control (ctrl) and mCherry-Zyxin (zyxin O/E) were transiently transfected into (**a**) D3 and (**b**) E14 cells with an increasing amount of plasmid DNA—0.1, 0.2, 0.5, 1.0, and 2.0 µg. The levels of pluripotency markers (Oct4 and Nanog) were analyzed by western blot. Zyxin overexpression was confirmed by detecting with both zyxin and mCherry antibodies. β-tubulin was used as loading control. **c** Densitometric analysis of pluripotency markers (Oct4 and Nanog) and zyxin was conducted and plotted for D3 and E14. Values were normalized against β-tubulin and presented as fold change using mCherry-control as the internal reference ($N = 3$). **d** Changes at transcript levels were analyzed through Real-Time PCR, using primers specifically targeting Oct4, Nanog and Zyxin. Relative quantification values were normalized against β-tubulin and presented as fold change using mCherry-control as the internal reference ($N = 3$). All results were from at least three independent experiments. Two-tailed unpaired Student's t-test was used to test the differences between the control and the highest concentration (2.0 µg) of mCherry-Zyxin (zyxin O/E). Error bars represent standard deviations. *$P ≤ 0.05$; **$P ≤ 0.01$; ****$P ≤ 0.0001$.

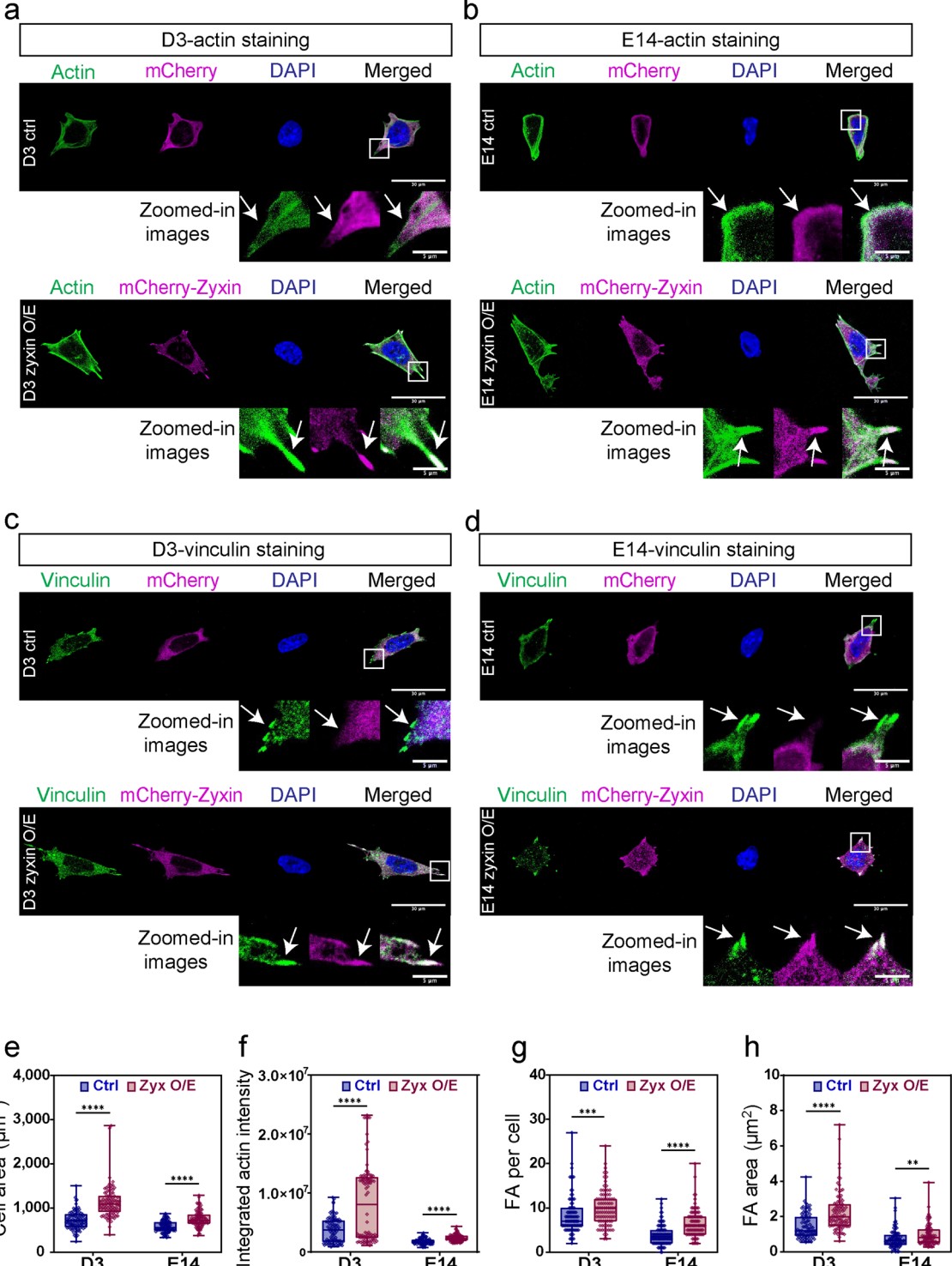

**Fig. 5 Zyxin overexpression increases abundance of f-actin and focal adhesions. a–d** To compare stress fibers and focal adhesions, D3/E14 transfected with mCherry vector control (ctrl) and mCherry-Zyxin (zyxin O/E) were immuno-stained with Phalloidin-Alexa 488 (**a**, **b**) and vinculin (**c**, **d**). White box areas were further amplified and shown as the zoomed in images. White arrows denoted f-actin and focal adhesion. Z-stacked images with maximum intensity projection were shown. Representative images from three biological repeats were shown. Scale bar: 30 μm for original images, 5 μm for zoomed in images. **e** projected cell area, **f** integrated actin intensity, **g** number of focal adhesions per cell, **h** projected focal adhesion areas were statistically analyzed and plotted. Quantified data were from three independent experiments ($N = 90$). Mann–Whitney $U$-test was used to test the differences between the control and zyxin overexpression groups. In the box and whisker plots, the center line is the median, the box-bounds are the 25th and 75th percentiles, and the whiskers are the 0.05 and 0.95 percentiles. **$**P \leq 0.01$; ***$P \leq 0.001$; ****$P \leq 0.0001$.

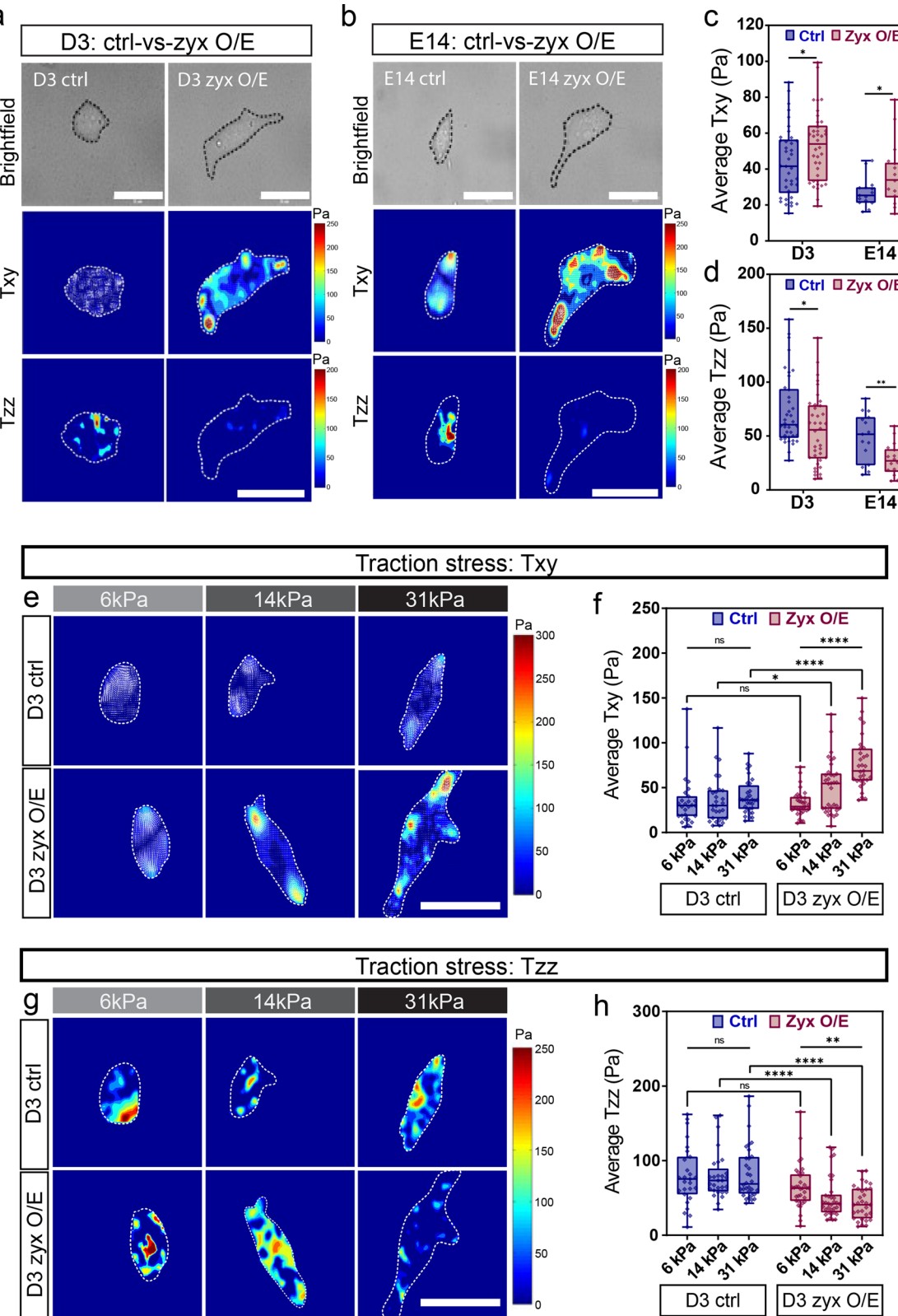

As it was previously found that the ECM type plays an important role in regulating the mechanotransduction and differentiation processes in stem cells[44], the above experiments were repeated using laminin coated surfaces to rule out the possibility of matrix specific effects. Consistent with previous findings using fibronectin, zyxin overexpressing D3 cells seeded on laminin were more sensitive to increasing substrate rigidities (Supplementary Fig. 7). Interestingly, zyxin overexpressing cells on laminin could sense rigidities better by increasing both Txy (Supplementary Fig. 7a, b) and Tzz (Supplementary Fig. 7c, d) stress magnitudes. Our data suggest that changes in ECM ligand type not only influence traction stress magnitudes as shown by previous studies[45,46], and could also impact on traction stress directions (both lateral and axial stresses).

**Fig. 6 Zyxin overexpression alters traction stress and facilitates substrate rigidity sensing. a** D3 and **b** E14 control cells and zyxin overexpression cells seeded on 14 kPa substrates were subjected to traction stress analysis. Brightfield images, in-plane traction stress images (Txy), out-plane traction stress images (Tzz) were compared between control and zyxin overexpression. **c**, **d** Average (**c**) in-plane (Txy) and (**d**) out-plane (Tzz) traction stress magnitudes were plotted for D3 (N = 38,38) and E14 (N = 15,18). Two-tailed unpaired Student's t-test was used to test the differences between the control and zyxin overexpression groups. **e** Traction stress maps for x-y direction (Txy) in increasing substrate rigidity of 6, 14, and 31 kPa were compared between control and zyxin overexpression. **f** Quantification analysis of average traction stress magnitudes in x-y direction (Txy) were compared between D3-control cells (D3 ctrl - Txy) (N = 27,29,32) and D3-zyxin overexpressing cells (D3 zyx O/E - Txy) (N = 33,34,31). Kruskal–Wallis test was used to test the differences among the three rigidities and Mann–Whitney U-test was used to test the differences between the control and zyxin overexpression groups at each rigidity. **g** Traction stress maps for z direction (Tzz) in increasing substrate rigidity of 6, 14, and 31 kPa were compared between control and zyxin overexpression. **h** Quantification analysis of average traction stress magnitudes in z direction (Tzz) were compared between D3-control cells (D3 ctrl - Tzz) (N = 27,29,32) and D3-zyxin overexpressing cells (D3 zyx O/E - Tzz) (N = 33,34,31). Kruskal–Wallis test was used to test the differences among the three rigidities and Mann–Whitney U-test was used to test the differences between the control and zyxin overexpression groups at each rigidity. In the box and whisker plots, the center line is the median, the box-bounds are the 25th and 75th percentiles, and the whiskers are the 0.05 and 0.95 percentiles. *$P \leq 0.05$; **$P \leq 0.01$; ****$P \leq 0.0001$; ns, not significant. Scale bar: 30 μm.

Taken together, zyxin-induced actin cytoskeletal and FA changes can modulate mESC cellular traction stress and result in increased sensitivity toward substrate rigidity. The very low overall zyxin levels in mESC may explain why mESC cannot sense substrate rigidity effectively.

**Zyxin regulates pluripotency genes through mechanotransduction of YAP.** Multiple signaling pathways are triggered by mechanosensing[41]. Hippo/YAP signaling is a well-studied mechanical stress-induced pathway. YAP has also been reported to regulate stem cell fate through transcriptional control of pluripotent gene expression[35,47]. To examine if zyxin overexpression affects YAP localization in response to increasing substrate rigidity, we monitored YAP distribution in the nucleus and cytosol of D3 control (Fig. 7a) and zyxin overexpressing D3 cells (Fig. 7b) grown on hydrogels with varying rigidity ranging from 6kPa, 14kPa to 31kPa. In D3 control cells, YAP was mainly localized in the nucleus across all substrate rigidities (Fig. 7a). When zyxin was overexpressed in D3 cells, YAP was mainly distributed in the nucleus on soft substrate (6 kPa) whereas a clear decrease in nuclear YAP was found on stiffer substrate (31 kPa). In contrast to control cells, the ratio of nuclear:cytosol YAP significantly decreased in response to increasing substrate rigidity in zyxin-overexpressing D3 cells (Fig. 7c). These observations were reproducible in E14 control versus zyxin-overexpressing E14 cell lines (Supplementary Fig. 6e, f). Collectively, our data suggest that YAP subcellular distribution is regulated by zyxin in a force-dependent manner.

It has been reported that phosphorylation of YAP at S127 by its upstream kinase in the Hippo pathway led to the inhibition of YAP activity and its retention in the cytoplasm through interaction with 14-3-3 protein[48]. We therefore analyzed the phosphorylation of YAP at this phospho-inhibitory site (S127). D3 control cells plated on increasing substrate rigidities displayed no significant changes in phospho-YAP to total YAP ratio. In contrast, in response to elevated substrate rigidity, phospho-YAP to total YAP ratio significantly increased in zyxin overexpressing cells (Fig. 7d, e). These observations were reproducible in E14 control versus zyxin-overexpressing E14 cell lines (Supplementary Fig. 6g, h). Therefore, we confirm that mechanotransduction through zyxin could modulate YAP activity. We then further examined if changes in YAP activity could affect pluripotent markers. Oct4 expression was assayed by western blot analysis in cells plated on substrates of different rigidity. We observed overall Oct4 reduction in zyxin overexpressing D3 (Fig. 7f) and E14 (Supplementary Fig. 6i) cells, which is consistent with our earlier findings showing that Oct4 is reduced upon zyxin overexpression (Fig. 4a, b), suggesting zyxin might play a role in regulating pluripotency through biochemical signaling. In addition, Oct4 level was further reduced in zyxin overexpressing cells on stiffer substrate (Fig. 7f, Supplementary Fig. 6i), suggesting zyxin-YAP-mediated mechanical signaling as

an additional and/or alternative regulation of Oct4 levels. In summary, we found that zyxin overexpression in mESC could increase rigidity sensing, which could lead to the inactivation of YAP and further impact on mESC pluripotency maintenance during early differentiation events.

## Discussion
Biochemical and mechanical cues are important in the regulation of stem cell fate[5,6]. Here, we report the crosstalk between mechanical signaling (traction stress and rigidity sensing) and biochemical signaling (YAP activity) mediated through focal adhesions in regulating stem cell fate (Fig. 7g). We find that the overall level of zyxin is very low in mESC with barely detectable presence at the FA (Fig. 2). Considering that zyxin is localized to the actin linkage and the force-transduction layers of mature FA[16,17], it is not surprising that the FAs found in mESC are immature and exhibit less traction stress and rigidity sensing. In this study, we found that zyxin expression is induced during early differentiation (Fig. 3) and FAs become more prominent as mESCs undergo differentiation. Zyxin overexpression in pluripotent cells positively influences the FAs and stress fibers (Fig. 5), which in turn modulate cellular traction stress to facilitate rigidity sensing (Fig. 6). In response to the improved rigidity sensing, YAP phosphorylation at S127 is increased, which leads to reduction of Oct4 levels (Fig. 7). Our previous study showed that in specialized cells, zyxin is involved in rigidity sensing and modulation of durotaxis[34]. Here, we propose that increased zyxin levels during mESC differentiation positively regulates actin cytoskeletal organization and traction stress exertion, which in turn modulate YAP activity.

An earlier study carried out in *Xenopus laevis* embryos and HEK293 cells also suggests that zyxin plays a significant role in regulating embryonic stem cell fate[49]. However, the proposed mechanisms differ from ours. In this earlier study, it is reported that zyxin negatively regulates pluripotency genes through inhibiting the binding of Ybx1 mRNA-stabilizing protein to Oct4 mRNA, resulting in the degradation of Oct4 mRNA[49]. The presence of nuclear zyxin may imply its participation in regulating gene expression[50,51]. Nevertheless, we found that overexpressed zyxin is primarily localized to the FA instead of the nucleus (Fig. 5c, d), suggesting that zyxin-mediated mechano-transduction might be another mechanism in pluripotency regulation. Surprisingly, a recent study reports that mice with zyxin depletion are viable and display no obvious developmental deficiency[52]. Perhaps, the existence of other zyxin family members, lipoma preferred partner (LPP)[53] and thyroid hormone receptor interactor 6 (Trip6)[54,55] may display functional redundancy and compensate for zyxin deficiency in in vivo development. LPP knockout mice also do not show significant developmental deficiency[56], while Trip6 knockout mice are not available yet.

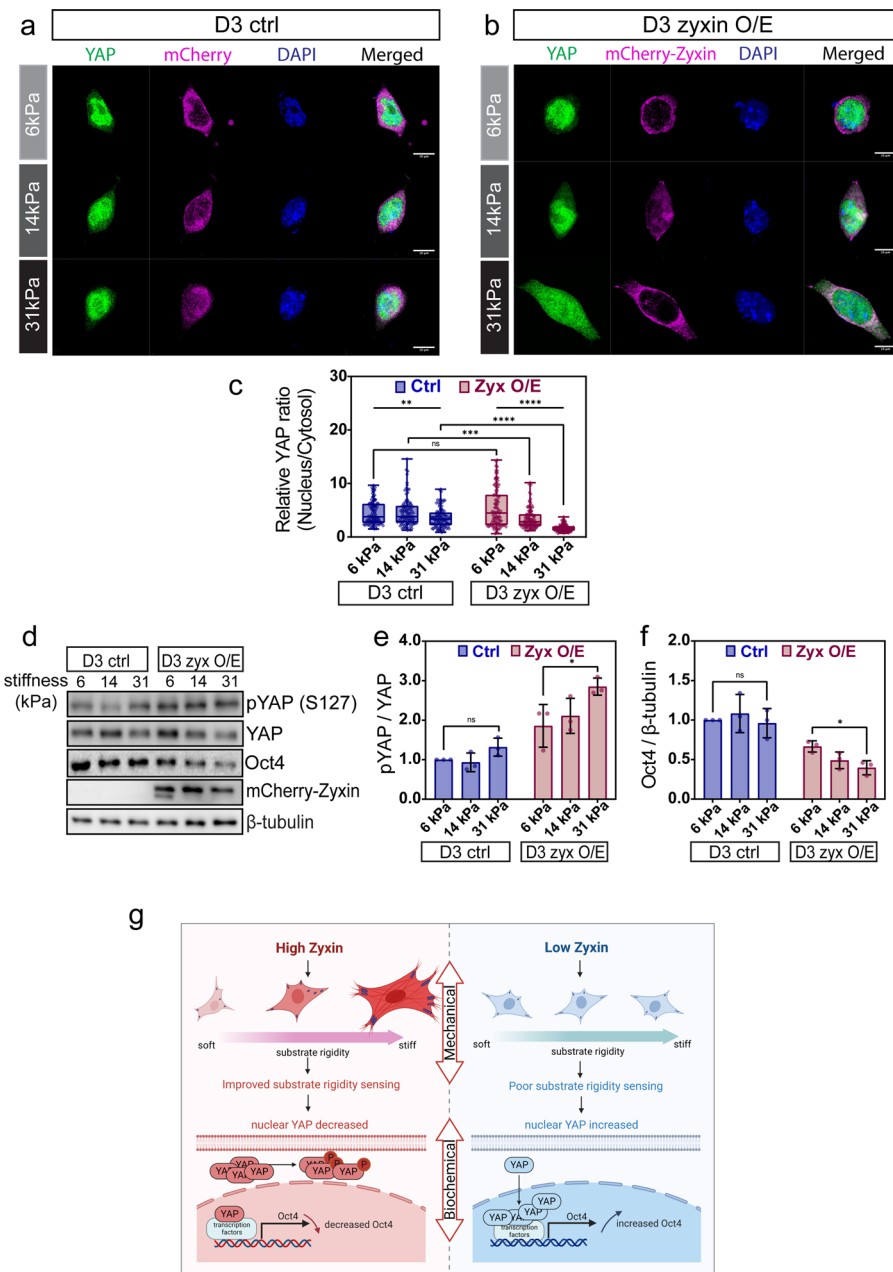

**Fig. 7 Zyxin overexpression deactivates YAP and downregulates Oct4 level in a force-dependent manner. a** D3 control and **b** D3-zyxin overexpressing cells were seeded on increasing substrate rigidity of 6, 14, and 31 kPa. YAP localization was examined by immunostaining. DAPI was used to observe cell nucleus. Z-stacked images with maximum intensity projection were shown. Representative images out of three biological repeats were presented. Scale bar: 10 μm. **c** Nucleus/Cytosol YAP ratio in D3 cells transfected with either control or mCherry-Zyxin plasmids were analyzed and plotted ($N = 100$). Kruskal–Wallis test was used to test the differences among the three rigidities and Mann–Whitney $U$-test was used to test the differences between the control and zyxin overexpression groups at each rigidity. In the box and whisker plots, the center line is the median, the box-bounds are the 25th and 75th percentiles, and the whiskers are the 0.05 and 0.95 percentiles. **d** D3 cells were transfected with either control or mCherry-Zyxin plasmids and then collected for western blot analysis using antibodies against pYAP, YAP, Oct4 and mCherry. β-tubulin was used as loading control. **e, f** Densitometry analysis of **e** pYAP (normalized against total YAP, $N = 3$) and **f** Oct4 (normalized against β-tubulin, $N = 3$). Two-tailed unpaired Student's $t$-test was used to test the differences between 6 and 31 kPa in each condition. Values presented as fold change using control cells plated on 6 kPa as the internal reference. Results were from three independent experiments. Error bars represent standard deviations. *$P \leq 0.05$; **$P \leq 0.01$; ***$P \leq 0.001$; ****$P \leq 0.0001$; ns, not significant. **g** Substrate rigidity sensing is improved in cells with high zyxin levels through the modulation of f-actin, focal adhesion and traction stress. As a result, there is increased phosphorylation of YAP (S127) and its retention in the cytosol, leading to lower nuclear YAP levels and reduced Oct4 expression (left panel). Substrate rigidity sensing is poor in cells with low zyxin levels. YAP is predominantly localized in the nucleus which can then promote Oct4 expression (right panel). Mechanical and biochemical cues work together through focal adhesions to regulate embryonic stem cells fate.

Further investigations are required to decipher the role of zyxin family proteins in early development.

We have shown that zyxin overexpression facilitates rigidity sensing, which could lead to YAP deactivation and potentially impact on pluripotent genes expression (Fig. 7). Our findings are in concordance with an earlier work conducted in the same mESCs[35], showing that YAP is inactivated in the differentiated mESC-D3 cells along with significant Oct4 reduction. Consistently, YAP knockdown leads to loss of pluripotency as indicated by Oct4 reduction, while constitutively active YAP mutant expression is sufficient to maintain pluripotency[35,36,57,58]. Nevertheless, more experiments are required to confirm the role of zyxin in the regulation of YAP phosphorylation and subsequent changes in Oct4 levels. Apart from LATS kinase involved in the canonical Hippo cascade[48], several other kinases such as Akt and NDR1/2 have been reported to phosphorylate YAP S127 leading to YAP cytoplasmic retention[59,60]. For future work, inhibiting these kinases to maintain YAP activation might be insightful to establish the mechanism of zyxin in the regulation of YAP and Oct4. In addition to zyxin overexpression experiments, it would be interesting to investigate zyxin knockdown effects on cell spreading and YAP localization/phosphorylation in mESCs. Apart from fibronectin and laminin, more studies can be conducted to compare mechanosensing, cell spreading, YAP localization and changes on pluripotency when mESCs are seeded on different ECMs, e.g., collagen and vitronectin. These comparisons are likely to provide more insights on the roles of ECM in regulating mESC mechanosensing and pluripotency.

We have shown that YAP's distribution in the nucleus decreased when zyxin overexpressing mESCs were plated on stiff substrates. Our observations contrast with a study carried out in human specialized cells and human mesenchymal stem cells (MSC) by Dupont et al., which shows that cells plated on stiff substrates promotes YAP nuclear localization in a Hippo-independent pathway but through RhoA-GTPases-mediated cytoskeleton tensions[61]. We observed a slight reduction of nuclear YAP in control cells on stiffer substrates (Fig. 7c), suggesting possible crosstalk with other signaling pathways in the regulation of YAP in addition to focal adhesion and cytoskeleton tensions. Our observations that phospho-YAP (Fig. 7e) increased on stiffer substrates and with synergistic effects from zyxin overexpression further support a role for zyxin in the regulation of YAP through the Hippo-LATS cascade[48]. Possible involvement of other kinases such as Akt and NDR1/2 has not escaped our attention[59,60]. In line with our observations in mESCs, it has been reported that YAP localization in the nucleus significantly decreased in differentiated D3 cells[35] and E14 cells[36] which exhibit more well-spread morphology. Our findings suggest that zyxin overexpressing mESCs resemble early differentiated cells. Therefore, apart from mechanotransduction, early differentiation is another factor to be considered in our current experimental system. The discrepancy might be explained through cell type differences in multi-levels of cell potency. Moreover, mESCs differ from hESCs in self-renewal and pluripotency maintenance mechanisms[62]. Therefore, more studies are needed to determine if our findings could be applied to the human stem cell system.

Till date, most studies on the roles of FAs in mechanobiology have used specialized cells as model systems[15]. Although FA architectures in mESCs[17] and hESCs[63] have been investigated recently, the function of individual focal adhesion component in ESCs remains relatively understudied. The role of talin in the assembly of focal adhesions in mESC has been reported, but whether talin is involved in regulating stemness remains to be explored[64]. Interestingly, FAK displays opposing roles in mESCs and hESCs[65]. While integrin signaling mediated through FAK promotes mESC differentiation towards endothelial cell lineage[66], FAK activation protects hESCs from apoptosis and differentiation[67]. In addition, although vinculin has been found to be involved in the regulation of multipotent stem cell (MSC) fate[68], it is not known whether it has a similar role in regulating ESCs. Our study on the specific role of zyxin in regulating ESC cell fate further bridges the gap of knowledge in individual roles of FA proteins. More work is needed to investigate how FA components work in synergy to regulate embryonic stem cell fate.

In summary, we find that zyxin is required for rigidity sensing (Yip et al.[34] and this study), which in turn stimulates YAP signaling to modulate pluripotent gene expression (Fig. 7g). Concerted mechanical and biochemical cues can work together through the FAs to regulate embryonic stem cell fate at early stages of differentiation. This crosstalk has important implications for stem cell fate decision.

## Methods

**Cell culture**. Mouse embryonic stem cells D3 (CRL-11632) and E14 (CRL-1821) were obtained from ATCC. Cells were plated on 0.1% gelatin-coated (Sigma-Aldrich, USA) dishes, and cultured in Dulbecco's Modified Eagle Medium (Thermo Fisher Scientific, USA) with 15% ES cell qualified fetal bovine serum (Invitrogen, USA), along with 1% non-essential amino acids (Invitrogen, USA), 1% penicillin-streptomycin (Invitrogen, USA), 1% sodium pyruvate (Invitrogen, USA), 0.2% β-mercaptoethanol (Invitrogen, USA) and 0.01% Leukemia Inhibitory Factor (LIF; Millipore, USA). Human embryonic stem cell (hESC) H1 and human induced pluripotent stem cell (hiPSC) with its parental cell (BJ-5ta) were gifts from Dr. Jonathan Yuin-Han Loh (A*STAR Institute of Molecular and Cell Biology, Singapore). Human pluripotent cells were cultured in feeder-free cell culture medium mTeSR[TM]1 (STEMCELL Technologies, Canada) according to the manufacturer's instructions. Mouse embryonic fibroblasts (MEF, ATCC, SCRC-1008) and NIH/3T3 mouse fibroblasts (ATCC, CRL-1658) and human fibroblast BJ-5ta cells were maintained in Dulbecco's Modified Eagle Medium (Thermo Fisher Scientific, USA) with 10% fetal bovine serum (Invitrogen, USA) and 1% penicillin-streptomycin (Invitrogen, USA). All cells were grown in 37 °C humidified incubator supplemented with 5% $CO_2$.

**Transfection**. The mCherry-Zyxin construct is from Dr. Pakorn Kanchanawong (Mechanobiology Institute, Singapore). Transfection was conducted using Lipofectamine 3000 (Thermo Fisher Scientific, USA) following the manufacturer's instructions. To generate zyxin overexpressing cells, mESCs were transiently transfected with increasing amounts of mCherry-Zyxin (0.1, 0.2, 0.5, 1.0, and 2.0 μg). Control cells were transiently transfected with 2.0 μg of mCherry-vector corresponding to the highest DNA concentration amongst mCherry-Zyxin transfection. Equal amounts of lipofectamine were applied to all transfections to exclude variation due to stress induced by transfection reagents. Cells were harvested 72 h after transfection followed by western blot analysis. Zyxin overexpression with the highest DNA amount (2.0 μg) was applied to subsequent focal adhesion, actin, rigidity sensing, YAP translocation experiments.

Zyxin knockdown experiments were performed based on our previous method[34]. Similar number of cells were seeded and the transient transfection was carried out with 75 nM Stealth RNAi[TM] siRNA targeting mouse zyxin (Invitrogen, siRNA ID: MSS238956). Zyxin knockdown efficiency was as reported in our previous work[34]. To induce differentiation, transfected cells were cultured in medium supplemented with decreasing concentrations of Retinoic Acid (RA; Sigma-Aldrich, USA) from 5 μM, 2 μM, to 1 μM. RA-induced differentiation was carried out for 24, 48, and 72 h, followed by examination on pluripotency marker.

**SDS-PAGE and western blot**. Cells were lysed with RIPA buffer (Thermo Fisher Scientific, USA) and denatured at 95 °C before being resolved on 8% or 10% SDS-PAGE. Separated proteins were then transferred to polyvinylidene fluoride (PVDF) membrane (Bio-Rad, USA) and blocked with 5% milk. The membrane was incubated with primary antibody overnight at 4 °C, followed by secondary antibody incubation for 1 h at room temperature. Chemiluminescent signal was detected with ChemiDoc[TM] MP imaging System (Bio-Rad, USA) using Amersham[TM]ECL[TM] Western Blotting Detection Reagents (GE Healthcare, UK). Image processing and band densitometry were done with Image Lab[TM] 6.0.1 software (Bio-Rad, USA). β-tubulin was used as loading control. All antibodies used were listed in Supplementary Table 1. Un-cropped western blots accompanied by size markers were presented in Supplementary Fig. 8.

**Immunostaining and image processing**. Similar number of cells were seeded on 10 μg/mL fibronectin-coated (Roche, USA) or 10 μg/mL laminin-coated (Sigma-Aldrich, USA) coverslips for 6 h to allow for single cell imaging. Samples were fixed with 4% formaldehyde (Sigma-Aldrich, USA) for 20 min at room temperature, permeabilized in 0.2% Triton-X-100 (Sigma-Aldrich, USA) for 10 min at room

temperature, and blocked with 4% bovine serum albumin (Sigma-Aldrich, USA) for minimum 30 min. Cells were then incubated with primary antibodies in blocking buffer at 4 °C overnight, followed by secondary antibody incubation for 1 h at room temperature. Stress fibers were stained with phalloidin conjugated with Alexa Fluor 546/488 (Invitrogen, USA). Coverslips were mounted using VECTASHIELD antifade mounting medium with DAPI (Vector Laboratories, USA). Images were acquired using Zeiss LSM 710/980 confocal microscopes (Carl Zeiss, Germany) and analyzed with FIJI software (NIH, version 1.53)[69]. Co-localization analysis was carried out with Coloc2 plugin in FIJI software (NIH, version 1.53c)[69] with the segmented FA as the mask[34].

**Fabrication of polyacrylamide (PAA) gel substrates**. Glass coverslips with diameter of 25 mm were first treated with silane solution, consisting of 1.2% 3-methacryloxypropyltrimethoxysilane (Shin-Etsu Chemical, Japan) and 2% acetic acid, (Schedelco, Singapore) for 2 h at room temperature to promote gel attachment on the glass coverslip. After silanization, the coverslips were washed with ethanol and air dried. Fabrication of polyacrylamide (PAA) gels was carried out by co-polymerization of bis-acrylamide (Bio-Rad, Hercules, CA) with N-acryloyl-6-aminocaproic acid (ACA; Tokyo Chemical Industry, Japan) to improve the stability of protein binding to PAA gels.

The ACA co-polymerized gels with Young's modulus of 6, 14, and 31 kPa were prepared on the silanized glass coverslips[34,70]. PAA gels were prepared with acrylamide (Bio-Rad, Hercules, CA) and Bis (Bio-Rad, Hercules, CA) with the addition of 100 mM ACA to obtain different rigidities of gel substrates. To visualize substrate deformation and calculate traction stress magnitudes exerted by the cells, red fluorescent beads 0.2 μm in diameter (Life Technologies, USA) were added to the gel solution. To initiate polymerization, ammonium persulfate (APS) (Sigma-Aldrich, USA) and tetramethylethylenediamine (TEMED) (Bio-Rad, Hercules, CA) were added to the gel solution, each with a final percentage of 0.01%. The silanized coverslip was mounted with 4 μL gel solution followed by covering with a 12 mm non-treated circular coverslip. The sandwiched gels were incubated for 10 min at 37 °C. The circular coverslip was carefully removed after gel polymerization. Then gels were fully hydrated in MES buffer [0.1 M 2-(N-morpholino) ethanesulfonic acid, 0.5 M sodium chloride, pH 6.1 (Sigma-Aldrich, USA)]. The hydrated gels with red fluorescent beads embedded were about 30–50 μm thick.

The gels are pre-coated with extracellular matrix protein to facilitate cell attachment to the PAA surface. A dehydration condensation reaction was carried out using water soluble carbodiimide to immobilize fibronectin or laminin on the surface of ACA-co-polymerized gels with embedded red fluorescent beads. The carboxyl groups of the ACA gels were first to be activated with 0.5 M N-hydroxysuccinimide (NHS, Wako Pure Chemical Industries, Osaka, Japan) and 0.2 M 1-ethyl-3-(3- dimethylaminopropyl) carbodiimide hydrochloride (EDAC, Dojindo Laboratories, Kumamoto, Japan) solution for 30 min at room temperature on a shaker. The gels were washed with 60% cold methanol diluted with phosphate-buffered saline (PBS; Gibco, USA) for 1 h at 4 °C. After that, the surfaces of ACA-co-polymerized gels were conjugated with 50 μg/mL fibronectin (Roche, USA) or 60 μg/mL laminin (Sigma-Aldrich, USA) at 4 °C overnight on a shaker. Finally, gels were transferred to 0.5 M ethanolamine (Sigma-Aldrich, USA) diluted by HEPES buffer [0.5 M HEPES (Sigma-Aldrich, USA), pH 9.0] for 30 min at 4 °C. The gels were washed twice with HEPES buffer at 4 °C, followed by three washes using PBS. Prior to cell seeding, the gels were sterilized with UV light for 15 min.

**Measurement for 3D traction stress**. A day prior to imaging, an equal number of cells were seeded on the polyacrylamide (PAA) gel substrates attached to the glass coverslips. Phase contrast images of the cells and fluorescence images of the beads were acquired using Zeiss Axio Observer Z1 inverted fluorescent microscope with EC Plan-NeoFluar, 100x oil objective lens (Numerical Aperture 1.3) (Carl Zeiss Microscopy, Germany). Z-stacks were obtained at interval of 0.1 μm, to enable the calculation of three-dimensional (3D) traction stress exerted by cells. The 3D traction stress calculation algorithm was applied to calculate traction stress magnitudes[39]. Two sets of the fluorescent beads imaging were acquired before and after trypsinization. The 3D displacement vectors were calculated by applying the digital volume correlation algorithm as developed by Franck et al.[38,71]. The gel strain tensor ε was determined based on the displacement-gradient technique. The material stress tensor σ can be obtained from the material's constitutive relation, $\sigma = E\varepsilon/(1 + v)$ [E is the Young's modulus of the gel and v is the Poisson's ratio of the gel ($v = 0.5$)]. The traction stress vectors $\vec{T}$ at the surface of the gel were calculated from the Cauchy relationship, $\vec{T} = \sigma \cdot \vec{n}$ ($\vec{n}$ is the surface normal vector). The stress magnitudes in x-y-direction ($\left|T_{xy}\right|$) and z-direction ($\left|T_{zz}\right|$) were calculated as given in Eqs. (1–2) respectively.

$$\left|T_{xy}\right| = \sqrt{\left|\vec{T}_x\right|^2 + \left|\vec{T}_y\right|^2} \tag{1}$$

$$\left|T_{zz}\right| = \sqrt{\left|\vec{T}_z\right|^2} \tag{2}$$

**Quantitative real-time PCR**. Total RNA was extracted from control and experimental cells, using the RNeasy mini kit (Qiagen, Germany). A total of 2 μg RNA was converted to cDNA using SuperScript VILO cDNA synthesis kit (Thermo

Fisher Scientific, USA). Real-time PCR was performed with SYBR Green real-time PCR master mix (Thermo Fisher Scientific, USA). Amplification was recorded and analyzed in the StepOne Plus real-time PCR system (Thermo Fisher Scientific, USA). All steps were performed according to the manufacturer's instructions. β-tubulin was used to normalize gene expression. Target gene expression was presented as fold-change to their respective controls. All primers used were listed in Supplementary Table 1.

**Cell area, focal adhesion, and actin intensity measurement**. Immunostained images were processed and analyzed using Fiji ImageJ software (version 1.53c)[69]. 3D image stacks were processed into 2D images through the maximum intensity projection algorithm. Cell area was then quantified by measuring the segmented region of interest (ROI) created through Otsu method thresholding[72] of the actin fiber channel. For actin intensity and focal adhesions, they were manually selected on the actin fiber channel and vinculin-stained channel respectively through Otsu method thresholding, and then added to the ROI. The whole cell ROI was also generated as above. The ROIs of the respective actin fibers and focal adhesions were then filled to obtain a segmented image. Actin intensity was quantified by measuring the cell ROI containing the segmented actin fibers while focal adhesion area and number were quantified using the "Analyze Particles" plugin on the cell ROI containing the segmented focal adhesions. All measurements were exported to GraphPad Prism version 9.4.1 for Windows, GraphPad Software, San Diego, California USA, www.graphpad.com, for analysis.

**Nuclear/cytosol YAP intensity measurement**. Image analysis of YAP nuclear:cytosolic localization ratio was analyzed using Fiji ImageJ software (version 1.53c)[69]. Single 2D images of individual immunostained cells were generated using the maximum intensity projection algorithm from the imaged Z-stacks. A nuclear and a cytosolic ROI was generated from the DAPI and YAP-stained channels respectively, through segmentation by Otsu method thresholding[72]. An area outside the cell on the YAP-stained channel was selected as ROI for background signal. The area and sum intensity in the ROIs were measured and exported for analysis. Background was subtracted from the nuclear and cytosolic sum intensities, then the nuclear intensity was divided by the cytosolic intensity to obtain the nuclear:cytosolic ratio of YAP.

**Statistics and reproducibility**. Statistical analysis was performed with the Graph-Pad Prism version 9.4.1 for Windows, GraphPad Software, San Diego, California USA, www.graphpad.com. All data was subject to the Shapiro-Wilk normality test before the respective statistical methods were carried out. Statistical significance between two groups was determined through two-tailed unpaired Student's t-test or Mann–Whitney U-test. Between three groups, it was determined through the one-way ANOVA or Kruskal-Wallis test. At least three independent experiments were analyzed. Sample sizes were described in the individual figure legend. P-values ≤ 0.05 were considered significant. *$P \leq 0.05$; **$P \leq 0.01$; ***$P \leq 0.001$; ****$P \leq 0.0001$; ns, not significant. Error bars represent standard deviations. In the box and whisker plots, the center line is the median, the box-bounds are the 25th and 75th percentiles, and the whiskers are the 0.05 and 0.95 percentiles.

**Reporting summary**. Further information on research design is available in the Nature Portfolio Reporting Summary linked to this article.

## Data availability
All data generated during the current study were included in this article and its Supplementary Information. Un-cropped western blots accompanied by size markers were presented in Supplementary Fig. 8. Description of Additional Supplementary Files was provided to describe Supplementary Data 1–13. Briefly, all source data underlying the graphs and charts were included in Supplementary Data 1–13.

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

## Acknowledgements

We thank Singapore Ministry of Education MOE Tier 2 grant (MOE2018-T2-1-058), MOE Tier 1 grant (RG106/20) and Nanyang Technological University, Singapore SUG grant to C.G.K. for funding support. We also thank Jonathan Loh Yuin-Han (Institute of Molecular and Cell Biology, ASTAR) for the gift of H1, hiPSC and BJ-5ta cell lines and Pakorn Kanchanawong (Mechanobiology Institute, Singapore) for sharing the mCherry-Zyxin cDNA construct.

## Author contributions

S.Z. designed and performed experiments, analyzed data and wrote the manuscript. L.H.C. performed experiments and analyzed data. J.Y.X.W. performed experiments and analyzed data. T.X.C. performed experiments and analyzed data. E.C. performed experiments. A.K.Y. designed and performed experiments. H.Y.L. designed experiments and analyzed data. K.H.C. designed the project and wrote the manuscript. C.G.K. designed the project and wrote the manuscript.

## Competing interests

The authors declare no competing interests.
