## [Peer Review File · Communications Biology]

Reviewers' comments:

Reviewer #1 (Remarks to the Author):

In this straightforward and well-written manuscript, the authors have shown the potential role of zyxin in the early stage of retinoic acid (RA)-induced mESC fibrogenic differentiation. The authors first show that zyxin is progressively upregulated within 3 days of differentiation by RA treatment. The role of zyxin in this process is confirmed by the overexpression of zyxin, which leads to cell spreading, increased traction forces, and suppressed pluripotency markers like Nanog and Oct4, resembling the RA-induced differentiation. Lastly, they showed that the effects of zyxin overexpression only occur in a stiffer 31 kPa environment, including the deactivation of YAP and the reduction of Oct4 level. This is a timely manuscript, contributing to ESC biology and the potential role of mechanical cues in ESC differentiation. Hence, after addressing the comments below, this manuscript should be considered for publication:

- Results, Fig. 4 shows that overexpression of zyxin leads to downregulation of pluripotency markers Nanog and Oct4. In order to confirm zyxin's role in RA-induced differentiation, further experimentation will be insightful, such as the knockdown of zyxin during RA-induced differentiation, i.e. will zyxin KD inhibits differentiation, and leads to the maintenance of the pluripotency markers level?
- Results/Discussion of Fig. 7, the findings in Fig.7 show the potential correlation between YAP deactivation and Oct4 reduction. The current findings are not enough to show the causative relationship between the two. Hence the statement in the Discussion section "In response to the enhanced sensing of rigidity, YAP's translocation to the nucleus is blocked, and YAP signaling-mediated pluripotency is impaired (Fig. 7)." is a bit too overarching. The author may alter the discussion section to more accurately represent the current finding. Or further experimentation is needed to confirm the role of YAP deactivation on Oct4 reduction. For example, if we KD/KO/inhibit LATS to maintain activation of YAP during the zyxin overexpression, will Oct4 level be maintained in this situation?
- Results/Discussion of Fig.7, the findings here are opposite to the previous findings in Hippo pathway mechanotransduction (Dupont et al. Nature 2011), where the spreading of multiple cell types leads to nuclear localization and activation of YAP. Please discuss the connection between this study and the findings in this manuscript.
- Discussion, "Biochemical and mechanical cues are important in the regulation of stem cell fate (5,6).", typo for 'regulation'.
- Figure 1E, and similar subsequent figure, for uniformity to the text, please change the y-axis to 'integrated actin intensity'.
- Figure 1E legend, colocalization of what?
- Figure 4A/B and similar subsequent figures, mCherry vector is the control, but it's negative for mCherry. Either change the label to mCherry-Zyxin, or please show the mCherry band for the control.
- Figure 6, what is the gel stiffness used for Figure 6A-D?
- Figure 6C and D, what does 'average ratio' mean? Please clarify either in the legend or text.
- Introduction, "Amongst the biophysical cues, traction stress, and substrate stiffness are found to profoundly influence stem cell status (7-9)", please include the relevant work by Engler, A et al. Cell 2006 on the role of matrix stiffness on MSC differentiation.
- Material and methods, Cell culture and transfection, "Human embryonic stem cells (hESC) H1 and human-induced embryonic stem cells (hiPSC) ...", is it supposed to be induced pluripotent stem cells?
- Material and methods, PAA fabrication, "In addition, red fluorescent beads 0.2 mm in diameter (Life Technologies ...", is mm supposed to be μm ?

Reviewer #2 (Remarks to the Author):

The manuscript by Zhang et al. describes the role of zyxin in early mouse embryonic stem cell differentiation. They report a low level of zyxin expression in mESCs and an absence of zyxin from focal adhesions despite the presence of other focal adhesion components. They then show that zyxin expression and localisation to focal adhesions occur when the cells are differentiated using

retinoic acid. Over expression of zyxin leads to a decline in pluripotency factor expression (Oct4 and Nanog), coinciding with larger focal adhesions and actin stress fibres. Furthermore, the authors detect that zyxin overexpression leads to increased mechanosensitivity.

Overall, the area of the study is intriguing, and this study provides new insights to the under-research area of FA mediated regulation of pluripotency. The answers to authors biological questions will be interesting to the mechano- and developmental biology community. Before the publishing this manuscript, major revision is, however needed.

Major comments:

1. Cells might alter their integrin expression during cell state changes. Altered integrin availability on cell membrane might lead to reduced adhesion and cell spreading on particular ECM. Reduced cell spreading and mechanosensing authors detect in their experiments might be a cause of this. The authors perform all their experiments on fibronectin-coated surfaces, which itself is interesting but might not give the complete picture. Currently, the authors cannot rule out the possibility that zyxin improves cell spreading and adhesion only on fibronectin. To address this, the authors should investigate the formation of focal adhesions, localisation of endogenous zyxin, and mechanosensitivity also on different ECM such as laminin.

2. Surprisingly the authors detect no or very low level of zyxin in FAs of undifferentiated mESCs and show a drastic rise of overall zyxin level and focal adhesion localisation upon differentiation (Figure 2 and 3). They, however, cite a paper (their ref. 16) where zyxin seem to be localised to FAs in IF staining in comparable conditions in undifferentiated mESCs. This might be due to fact that in this manuscript, the mESCs are compared to highly specified fibroblast cells where the zyxin expression is probably extreme. However, the authors should comment on this discrepancy.

3. In order to study the role of zyxin in the regulation of pluripotency, the authors overexpress zyxin in mESCs. Excitingly this declines the level of core pluripotency factors Oct4 and Nanog. However, the overexpression experiment is not adequately described in the methods section of the manuscript. For example, is the expression level of zyxin controlled by titrating the level of plasmid DNA, how long was the construct expressed before sample collection? Therefore, the authors need to describe the experiment according to journal standards. Furthermore, more controls are needed. 1) If the DNA amount used for transfections was titrated, the authors should also express the corresponding amount of control DNA to ensure that the decline of Oct4 and Nanog is not due to stress caused by overexpression. 2) The authors should show a microscope image of a cell expressing the highest level of zyxin used in the overexpression panel (Figure 4) to ensure that cells remain healthy.

4. The results of overexpression of zyxin are exciting, but the role of zyxin in the regulation mESC lineage specification remains elusive. In their abstract, the authors state that zyxin plays an important role in early lineage specification. This claim is not, however, experimentally proven. Furthermore, the manuscript would greatly benefit from an experiment where zyxin would be knocked down or knocked out during mESC differentiation. For example, this experiment could be coupled to an embryonic body assay where lineage-specific factors could be analysed and compared to control cells. Either experiments investigating lineage specification need to be conducted or claims of this need to be toned down.

5. In figure 7, the authors investigate the localisation of YAP during zyxin overexpression. There are, however few issues:

a. Authors state that Yap becomes predominantly cytosolic in a stiff environment when zyxin is overexpressed. This is not however supported by the IF image presented by the authors and the claim should be toned down or a more representative image shown.

b. The YAP N/C ratio quantification strategy needs to be described more accurately in the methods section. In the current form, it is not clear if mean or sum intensity was used for the quantification. This is important when comparing spread and unspread cells.

c. Analysing only 30 cells for this experiment feels very low for this kind of experiment. The n number should be increased.

- d. Yap seems to move out from the nuclei in a stiff environment in zyxin expressing cells. This is surprising, and the authors should discuss this.
- e. If Yap controls the Oct4 expression downstream of zyxin, it is surprising to see that Oct4 levels decline in the 6 kPa environment in zyxin expressing cells (Figure 7f.). The authors need to explain this or take this in consideration in their model (7g).

6. The authors present most of the data as bar graphs with SEM error bars. This does not give the complete picture of the data distribution. Instead, individual data points should be shown, and the variability of the data should be presented as a standard deviation. If SEM is used, it should be justified.

Minor comments:

1. The use of a red-green colour combination should be avoided when presenting the microscopy data.

2. The authors should state in the figures or in the legends the imaging plane and possible image projections, environment stiffnesses (Figure 6 c and d) and the statistical test used.

3. Authors should make sure that the methods section is up to date. For example, a detailed description of focal adhesion analysis, overexpression experiments and fluorescence intensity measurement is missing.

4. Authors should adjust their use of the following words in the text and figures and aim for accurate description:

- a. Size vs area. For example, when measuring the cell area from a single plane or projected images does not give the size of the cell.
- b. Cortex. When describing the localisation of actin fibres.
- c. Enhance. When describing the growth of focal adhesions or abundance of F-actin fibres.

Reviewers' comments:

Reviewer #1 (remarks to the author)

In this straightforward and well-written manuscript, the authors have shown the potential role of zyxin in the early stage of retinoic acid (RA)-induced mESC fibrogenic differentiation. The authors first show that zyxin is progressively upregulated within 3 days of differentiation by RA treatment. The role of zyxin in this process is confirmed by the overexpression of zyxin, which leads to cell spreading, increased traction forces, and suppressed pluripotency markers like Nanog and Oct4, resembling the RA-induced differentiation. Lastly, they showed that the effects of zyxin overexpression only occur in a stiffer 31 kPa environment, including the deactivation of YAP and the reduction of Oct4 level. This is a timely manuscript, contributing to ESC biology and the potential role of mechanical cues in ESC differentiation. Hence, after addressing the comments below, this manuscript should be considered for publication:

1. Results, Fig. 4 shows that overexpression of zyxin leads to downregulation of pluripotency markers Nanog and Oct4. In order to confirm zyxin's role in RA-induced differentiation, further experimentation will be insightful, such as the knockdown of zyxin during RA-induced differentiation, i.e. will zyxin KD inhibits differentiation, and leads to the maintenance of the pluripotency markers level?

We thank the reviewer for the suggestion. We have performed additional experiments to confirm zyxin's role in RA-induced differentiation. Our results suggest that zyxin knockdown could better maintain Oct4 levels under RA treatment compared to control. The results were presented in **new Fig. S3** and described in **Results section (Line 163 - Line 173)**:

“The observations that zyxin overexpression leads to reduction of pluripotent markers (Fig. 4a-d) prompted us to examine if zyxin knockdown could maintain pluripotency under RA-induced differentiation. Zyxin knockdown was achieved by transient transfection of siRNA with similar approach and knockdown efficiency reported in our previous work³⁴. Consistently, we observed induction of zyxin expression in cells treated with RA but not in cells transfected with siRNA (Fig. S3). Taking reference from previous studies^{35,36}, we next proceeded to titrate RA treatment from 5 μ M, 2 μ M, to 1 μ M in order to better observe the effects of zyxin knockdown. Oct4 levels were monitored over a 72-hour time course in response to zyxin knockdown. While RA-treatment led to significant reduction of Oct4 in control cells, Oct4 levels were better maintained in RA-treated zyxin-knockdown cells (Fig. S3). These results suggest a role of zyxin in pluripotency maintenance.”

Details of zyxin overexpression and knockdown experiments were described in the **Methods section (Line 411 - Line 431)**.

2. Results/Discussion of Fig. 7, the findings in Fig.7 show the potential correlation between YAP deactivation and Oct4 reduction. The current findings are not enough to show the causative relationship between the two. Hence the statement in the Discussion section “In response to the enhanced sensing of rigidity, YAP’s translocation to the nucleus is blocked, and YAP signaling-mediated pluripotency is impaired (Fig. 7).” is a bit too overarching. The author may alter the discussion section to more accurately represent the current finding. Or further experimentation is needed to confirm the role of YAP deactivation on Oct4 reduction. For example, if we KD/KO/inhibit LATS to maintain activation of YAP during the zyxin overexpression, will Oct4 level be maintained in this situation?

We agree with the reviewer that the current data only shows potential correlation between YAP deactivation and Oct4 reduction. We have revised the discussion to describe our current findings more accurately. We included discussion on previous studies describing positive correlation between YAP deactivation and Oct4 reduction. Possible involvement of LATS and other potential YAP kinases were proposed for future work in the **Discussion section (Line 335 – Line 346)**:

“We have shown that zyxin overexpression facilitates rigidity sensing, which could lead to YAP deactivation and potentially impact on pluripotent genes expression (Fig. 7). Our findings are in concordance with an earlier work conducted in the same mESCs³⁵, showing that YAP is inactivated in the differentiated mESC-D3 cells along with significant Oct4 reduction. Consistently, YAP knockdown leads to loss of pluripotency as indicated by Oct4 reduction, while constitutively active YAP mutant expression is sufficient to maintain pluripotency^{35,36,57,58}. Nevertheless, more experiments are required to confirm the role of zyxin in the regulation of YAP phosphorylation and subsequent changes in Oct4 levels. Apart from LATS kinase involved in the canonical Hippo cascade⁴⁸, several other kinases such as Akt and NDR1/2 have been reported to phosphorylate YAP S127 leading to YAP cytoplasmic retention^{59,60}. For future work, inhibiting these kinases to maintain YAP activation might be insightful to establish the mechanism of zyxin in the regulation of YAP and Oct 4.”

3. Results/Discussion of Fig.7, the findings here are opposite to the previous findings in Hippo pathway mechanotransduction (Dupont et al. Nature 2011), where the spreading of multiple cell types leads to nuclear localization and activation of YAP. Please discuss the connection between this study and the findings in this manuscript.

Dupont and his co-workers (Dupont et al., 2011) were the first group to report that cells spreading on stiff substrates promotes YAP nuclear localization which is independent of Hippo cascade but requires Rho GTPases and cytoskeleton tensions. We have acknowledged the significance of this work in linking rigidity sensing and YAP activity. The **Discussion section** has been substantially re-worked to compare the consistency and discrepancy between our findings with Dupont’s study carried out in human specialized cells and human mesenchymal stem cells, as well as other studies carried out in mouse and human embryonic stem cells (**Line 348 – Line 367**):

“We have shown that YAP’s distribution in the nucleus decreased when zyxin overexpressing mESCs were plated on stiff substrates. Our observations contrast with a study carried out in human specialized cells and human mesenchymal stem cells (MSC) by Dupont et al., which shows that cells plated on stiff substrates promotes YAP nuclear localization in a Hippo-

independent pathway but through RhoA-GTPases-mediated cytoskeleton tensions⁶¹. We observed a slight reduction of nuclear YAP in control cells on stiffer substrates (Fig. 7c-control), suggesting possible crosstalk with other signaling pathways in the regulation of YAP in addition to focal adhesion and cytoskeleton tensions. Our observations that phospho-YAP (Fig. 7e) increased on stiffer substrates and with synergistic effects with zyxin overexpression further support a role for zyxin in the regulation of YAP through the Hippo-LATS cascade⁴⁸. Possible involvement of other kinases such as Akt and NDR1/2 has not escaped our attention^{59,60}. In line with our observations in mESCs, it has been reported that YAP localization in the nucleus significantly decreased in differentiated D3 cells³⁵ and E14 cells³⁶ which exhibit more well-spread morphology. Our findings suggest that zyxin overexpressing mESCs resemble early differentiated cells. Therefore, apart from mechanotransduction, early differentiation is another factor to be considered in our current experimental system. The discrepancy between our observations and those reported by Dupont et al., might be explained through cell type differences in multi-levels of cell potency. Moreover, mESCs differ from hESCs in self-renewal and pluripotency maintenance mechanisms⁶². Therefore, more studies are needed to determine if our findings could be applied to the human stem cell system.”

4. Discussion, “Biochemical and mechanical cues are important in the regulation of stem cell fate (5,6).”, typo for ‘regulation’.

We thank reviewer for the correction. Typo for “regulation” was corrected and shown in **Discussion section (Line 302)**:

“Biochemical and mechanical cues are important in the regulation of stem cell fate.”

5. Figure 1E, and similar subsequent figure, for uniformity to the text, please change the y-axis to ‘integrated actin intensity’.

We have changed the y-axis to “integrated actin intensity” for all similar figures and figure legends. Changes were reflected in the following figures:

Fig. 1e

Fig. 5f

Fig. S4f

6. Figure 3E legend, colocalization of what?

We apologize for the unclear description. We have described as “Classical colocalization analysis of zyxin and vinculin” in **Fig. 3e legend (Line 823 – Line 824)**:

“Fig. 3e: Classical colocalization analysis of zyxin and vinculin was conducted from three independent experiments (N=60).”

7. Figure 4A/B and similar subsequent figures, mCherry vector is the control, but it's negative for mCherry. Either change the label to mCherry-Zyxin, or please show the mCherry band for the control.

We thank reviewer for the correction. We have changed the label to mCherry-Zyxin for all similar figures and figure legends. Changes were reflected in the following figures:

Fig. 4a-b
Fig. 5a-d
Fig. 7b and d
Fig. S4a-d
Fig. S6g and i

8. Figure 6, what is the gel stiffness used for Figure 6A-D?

The stiffness used for Fig. 6a-d is 14 kPa. We have included the detail into **Fig. 6 legend (Line 857 – Line 858)**:

“Fig. 6: a D3 and b E14 control cells and zyxin overexpression cells seeded on 14 kPa were subjected to traction stress analysis.”

9. Figure 6C and D, what does ‘average ratio’ mean? Please clarify either in the legend or text.

We meant the average traction stress magnitudes. We have modified **Fig. 6 legend (Line 860 – Line 861)**:

“Fig. 6: Average (c) in-plane (T_{xy}) and (d) out-plane (T_{zz}) traction stress magnitudes were plotted for D3 ($N=38,38$) and E14 ($N=15,18$).”

10. Introduction, “Amongst the biophysical cues, traction stress, and substrate stiffness are found to profoundly influence stem cell status (7–9)”, please include the relevant work by Engler, A et al. Cell 2006 on the role of matrix stiffness on MSC differentiation.

Matrix elasticity was first identified as a new factor in regulating stem cell fate by Discher’s group in 2006 (Engler et al., 2006). We have acknowledged this important study and included citation in the **Introduction section (Line 36 - Line 37)**:

“Amongst the biophysical cues, traction stress and substrate stiffness are found to profoundly influence stem cell status^{7–10}.”

11. Material and methods, Cell culture and transfection, “Human embryonic stem cells (hESC) H1 and human-induced embryonic stem cells (hiPSC) ...”, is it supposed to be induced pluripotent stem cells?

We apologize for the mistake. We have corrected the text as shown in **Methods section (Line 401 – Line 402)**:

“Human embryonic stem cells (hESC) H1 and human induced pluripotent stem cells (hiPSC).....”

12. Material and methods, PAA fabrication, “In addition, red fluorescent beads 0.2 mm in diameter (Life Technologies ...”, is mm supposed to be μm ?

We apologize for the mistake. We have corrected the text as shown in **Methods section (Line 475)**:

“Red fluorescent beads 0.2 μm in diameter (Life Technologies, Singapore) were added to the gel solution to allow visualization of substrate deformation and calculation of traction stresses exerted by the cell.”

Reviewer #2 (Remarks to the Author):

The manuscript by Zhang et al. describes the role of zyxin in early mouse embryonic stem cell differentiation. They report a low level of zyxin expression in mESCs and an absence of zyxin from focal adhesions despite the presence of other focal adhesion components. They then show that zyxin expression and localisation to focal adhesions occur when the cells are differentiated using retinoic acid. Over expression of zyxin leads to a decline in pluripotency factor expression (Oct4 and Nanog), coinciding with larger focal adhesions and actin stress fibres. Furthermore, the authors detect that zyxin overexpression leads to increased mechanosensitivity.

Overall, the area of the study is intriguing, and this study provides new insights to the under-research area of FA mediated regulation of pluripotency. The answers to authors biological questions will be interesting to the mechano- and developmental biology community. Before the publishing this manuscript, major revision is, however needed.

Major comments:

1. Cells might alter their integrin expression during cell state changes. Altered integrin availability on cell membrane might lead to reduced adhesion and cell spreading on particular ECM. Reduced cell spreading and mechanosensing authors detect in their experiments might be a cause of this. The authors perform all their experiments on fibronectin-coated surfaces, which itself is interesting but might not give the complete picture. Currently, the authors cannot rule out the possibility that zyxin improves cell spreading and adhesion only on fibronectin. To address this, the authors should investigate the formation of focal adhesions, localisation of endogenous zyxin, and mechanosensitivity also on different ECM such as laminin.

We agree with the reviewer that investigation using only one type of matrix molecule (fibronectin) cannot rule out the possibility of matrix specific effects. To address this issue and to reinforce roles of zyxin in regulating focal adhesion, experiments in Fig. 5 were repeated using a different matrix, laminin. Similar results were obtained when cells were plated on laminin-coated surfaces. Our observations are consistent with Xia et al. (Xia et al., 2019) who showed that mESCs exhibited similar pluripotency, spreading, actin and focal adhesions when plated on three different matrixes: fibronectin, laminin, and gelatin. Additional experiments performed were presented in **new Fig. S4** and described in **Results section (Line 189 -Line 195)**:

“To rule out the possibility of matrix specific effects, the above experiments were repeated using a different matrix, laminin. We obtained similar results when cells were plated on laminin. mCherry-Zyxin consistently localized to FAs in the overexpression cells (Fig. S4c-d). Zyxin overexpressing D3 and E14 cells were more well spread and exhibited increased actin staining as well as FA numbers and areas (Fig. S4). Together, we found that high zyxin level down-regulates pluripotency markers (Fig. 4) and concurrently increases actin and FAs (Fig. 5 and Fig. S4) in pluripotent cells.”

Moreover, to reinforce the roles of zyxin in regulating rigidity sensing, experiments described in Fig. 6 were repeated using a different matrix, laminin. Consistently, zyxin overexpressing cells exhibited improved rigidity sensing on both fibronectin and laminin coated surfaces.

Mechanosensing on laminin were presented in **new Fig. S7** and described in **Results section (Line 248 – Line 256)**:

“As it was previously found that the ECM type plays an important role in regulating the mechanotransduction and differentiation processes in stem cells⁴⁴, the above experiments were repeated using laminin coated surfaces to rule out the possibility of matrix specific effects. Consistent with previous findings using fibronectin, zyxin overexpressing D3 cells seeded on laminin were more sensitive to increasing substrate rigidities (Fig. S7c and f). Interestingly, zyxin overexpressing cells on laminin could sense rigidities better by increasing both T_{xy} (Fig. S7a-c) and T_{zz} (Fig. S7d-f) stress magnitudes. Our data suggest that changes in ECM ligand type not only influence traction stress magnitudes as shown by previous studies^{45,46}, but could also impact on traction stress directions (both lateral and axial stresses).”

2. Surprisingly the authors detect no or very low level of zyxin in FAs of undifferentiated mESCs and show a drastic rise of overall zyxin level and focal adhesion localisation upon differentiation (Figure 2 and 3). They, however, cite a paper (their ref. 16) where zyxin seem to be localised to FAs in IF staining in comparable conditions in undifferentiated mESCs. This might be due to fact that in this manuscript, the mESCs are compared to highly specified fibroblast cells where the zyxin expression is probably extreme. However, the authors should comment on this discrepancy.

We agree with the reviewer that the extremely high zyxin expression in MEF may affect detection of very low levels zyxin in mESCs. To improve detection of low abundant zyxin in mESC, we repeated the western blot (1) without MEF protein lysate in the same blot, (2) increased the amount of total protein loaded in the wells and (3) used highly sensitive chemiluminescent substrates. The optimized western blot analysis confirmed the presence of endogenous zyxin in mESCs although at very low levels which could be induced by RA treatment (**new Fig. S2**). Consistently, zyxin transcript could be detected at low levels through RT-PCR in mESCs cells (Fig. 2b). Taken together, our results showed that zyxin is present at very low levels in pluripotent mESCs (Fig. S2 and Fig. 2b) and the levels could be significantly induced in the differentiated cells (Fig. 3a-b). Xia et al. also reported that LIM-domain containing FA proteins were less abundant and this observation appeared to be a common signature for mESCs and hESCs (Xia et al., 2019). Xia et al. showed endogenous zyxin localization to focal adhesions and concluded that mESCs are capable of forming mature FAs (Xia et al., 2019). However, we were not able to observe this in our imaging experiments which could suggest that the FAs formed in mESC are less mature compared to adherent cell types. We now include **new Fig. S2** to show presence of zyxin at very low levels in mESC and that treatment with RA could induce the expression of zyxin. We have revised our **Results section** to describe our findings more accurately (**Line 124 – Line 146**):

“We found that the levels of zyxin were consistently much lower in mESCs and hESCs compared with their respective differentiated cell types (Fig. 2a). In particular, zyxin was found at very low levels in mESCs (Fig. 2a and Fig. S2a). This is consistent with an earlier study which reported that LIM-domain containing FA proteins are lower in abundance in mESCs¹⁷. Real-time PCR was adopted to verify the mRNA expression of zyxin in mESC. Data from two independent sets of zyxin primers consistently showed that zyxin transcript could be detected in mESCs but was ~1000-fold lower than that in embryonic fibroblasts (Fig. 2b). Due to its low abundance, we are thus not able to detect the presence of endogenous zyxin at the FA of

mESCs, whereas clear zyxin localization at the FA was observed in fibroblasts (Fig. 2c-d). Xia et al. showed endogenous zyxin localization to focal adhesions and concluded that mESCs are capable of forming mature FAs¹⁷. However, we were not able to observe this in our imaging experiments which could suggest that the FAs formed in mESC are less mature compared to adherent cell types.

We went on to examine FA protein expression profiles during RA-induced early differentiation. D3 and E14 cells were treated with RA over a 72-hour time course, and changes in core FA proteins were analyzed by western blot. Zyxin levels increased drastically when D3 cells was treated with RA (Fig. 3a-b and Fig. S2a-b). We also began to detect zyxin localization to the FA in RA-treated cells (Fig. 3c-d). Co-localization of zyxin and vinculin was observed at the FA of RA-treated cells (Fig. 3e). Earlier studies have found that mechanical force-induced actin polymerization at FA promotes zyxin accumulation at the adhesion area^{20,33}. Therefore, the increased zyxin accumulation at FA observed in early differentiated cells could suggest possible roles for zyxin in mechanotransduction and regulation of stem cell fate.”

3. In order to study the role of zyxin in the regulation of pluripotency, the authors overexpress zyxin in mESCs. Excitingly this declines the level of core pluripotency factors Oct4 and Nanog. However, the overexpression experiment is not adequately described in the methods section of the manuscript. For example, is the expression level of zyxin controlled by titrating the level of plasmid DNA, how long was the construct expressed before sample collection? Therefore, the authors need to describe the experiment according to journal standards. Furthermore, more controls are needed. 1) If the DNA amount used for transfections was titrated, the authors should also express the corresponding amount of control DNA to ensure that the decline of Oct4 and Nanog is not due to stress caused by overexpression. 2) The authors should show a microscope image of a cell expressing the highest level of zyxin used in the overexpression panel (Figure 4) to ensure that cells remain healthy.

We apologize for the missing experimental details. Zyxin was overexpressed at increasing amount from 0.1 μg , 0.2 μg , 0.5 μg , 1.0 μg to 2.0 μg . The constructs were expressed for 72 hours followed by sample collection. To exclude effects from transfection-induced cellular stress, control cells were transiently transfected with 2.0 μg of mCherry-empty vector corresponding to the highest DNA concentration amongst mCherry-zyxin transfection. Equal amount of lipofectamine was applied to all transfections to exclude stress induced by transfection reagents. We are certain that cells expressing 2.0 μg of mCherry-zyxin are healthy, because zyxin overexpression with the highest DNA amount (2.0 μg) was applied to all the following focal adhesion and actin immunostaining (Fig. 5 and Fig. S4), rigidity sensing (Fig. 6 and Fig. S6-7), and YAP translocation (Fig. 7 and Fig. S6) experiments. We have clearly described zyxin overexpression experiments in the **Methods section (Line 413-Line 423)**:

“The mCherry-Zyxin construct is from Dr. Pakorn Kanchanawong (Mechanobiology Institute, Singapore). Transfection was carried out using Lipofectamine 3000 (Thermo Fisher Scientific, USA) according to the manufacturer’s instructions. To generate zyxin overexpressing cells, mESCs were transiently transfected with increasing amounts of mCherry-Zyxin (0.1 μg , 0.2 μg , 0.5 μg , 1.0 μg , 2.0 μg). Control cells were transiently transfected with 2.0 μg of mCherry-vector corresponding to the highest DNA concentration amongst mCherry-Zyxin transfection. Equal amounts of lipofectamine were applied to all transfections to exclude variation due to stress induced by transfection reagents. Cells were harvested 72 hours after transfection

followed by western blot analysis. Zyxin overexpression with the highest DNA amount (2.0 µg) was applied to subsequent focal adhesion, actin, rigidity sensing, YAP translocation experiments.”

4. The results of overexpression of zyxin are exciting, but the role of zyxin in the regulation mESC lineage specification remains elusive. In their abstract, the authors state that zyxin plays an important role in early lineage specification. This claim is not, however, experimentally proven. Furthermore, the manuscript would greatly benefit from an experiment where zyxin would be knocked down or knocked out during mESC differentiation. For example, this experiment could be coupled to an embryonic body assay where lineage-specific factors could be analysed and compared to control cells. Either experiments investigating lineage specification need to be conducted or claims of this need to be toned down.

We agree with the reviewer that our current experiments only suggest the possible role of zyxin in regulating mESCs pluripotency. More experiments are required to establish the exact role of zyxin in early lineage specification. We have revised the **Abstract section** to describe our findings more accurately based on our current experiments (**Line 21 – Line 23**):

“Using an integrative biochemical and biophysical approach, we demonstrate involvement of zyxin in regulating pluripotency through actin stress fibres and focal adhesions which are known to modulate cellular traction stress and facilitate substrate rigidity-sensing.”

In addition, we have performed zyxin knockdown experiments as suggested by Reviewer 2 to confirm zyxin’s role in regulating mESC pluripotency. Our results suggest mESCs with zyxin-knockdown could still maintain Oct4 levels under RA treatment. The results were presented in new **Fig. S3** and described in **Results section (Line 163 - Line 173)**:

“The observations that zyxin overexpression leads to reduction of pluripotent markers (Fig. 4a-d) prompted us to examine if zyxin knockdown could maintain pluripotency under RA-induced differentiation. Zyxin knockdown was achieved by transient transfection of siRNA with similar approach and knockdown efficiency reported in our previous work³⁴. Consistently, we observed induction of zyxin expression in cells treated with RA but not in cells transfected with siRNA (Fig. S3). Taking reference from previous studies^{35,36}, we next proceeded to titrate RA treatment from 5 µM, 2 µM, to 1 µM in order to better observe the effects of zyxin knockdown. Oct4 levels were monitored over a 72-hour time course in response to zyxin knockdown. While RA-treatment led to significant reduction of Oct4 in control cells, Oct4 levels were better maintained in RA-treated zyxin-knockdown cells (Fig. S3). These results suggest a role of zyxin in pluripotency maintenance.”

Details of zyxin knockdown experiments were described in the **Methods section (Line 425 - Line 431)**:

“Zyxin knockdown experiments were performed based on our previous method³⁴. Briefly, similar number of cells were seeded and transiently transfected with 75 nM Stealth RNAi™ siRNA targeting mouse zyxin (Invitrogen, siRNA ID: MSS238956). Zyxin knockdown efficiency was as reported in our previous work³⁴. To initiate differentiation, transfected cells were cultured in medium supplemented with decreasing concentrations of Retinoic Acid (RA; Sigma-

Aldrich, USA) from 5 μ M, 2 μ M, to 1 μ M. RA-induced differentiation was carried out for 24 hours, 48 hours and 72 hours, followed by examination on pluripotency marker.”

5. In figure 7, the authors investigate the localisation of YAP during zyxin overexpression. There are, however few issues:

a. Authors state that Yap becomes predominantly cytosolic in a stiff environment when zyxin is overexpressed. This is not however supported by the IF image presented by the authors and the claim should be toned down or a more representative image shown.

We have revised the **Results section** to describe IF images clearly. We observed reduction of nuclear YAP in zyxin overexpressing cells on stiff substrate (**Line 271 – Line 275**):

“When zyxin was overexpressed in D3 cells, YAP was mainly distributed in the nucleus on soft substrate (6kPa) whereas a clear decrease in nuclear YAP was found on stiff substrate (31kPa). In contrast to control cells, the ratio of nuclear:cytosol YAP significantly decreased in response to increasing substrate rigidity in zyxin-overexpressing D3 cells (Fig. 7c).”

b. The YAP N/C ratio quantification strategy needs to be described more accurately in the methods section. In the current form, it is not clear if mean or sum intensity was used for the quantification. This is important when comparing spread and unspread cells.

We apologize for the missing details. It was mentioned that the “sum of pixel values in the ROIs were measured”. We have clarified in the **Methods section** that we use sum intensity for the quantification (**Line 555 – Line 565**):

“Nuclear/cytosol YAP intensity measurement

Image analysis of YAP nuclear:cytosolic localization ratio was analyzed using Fiji ImageJ software (version 1.53c)⁶⁹. Single 2D images of individual immunostained cells were generated using the maximum intensity projection algorithm from the imaged Z-stacks. A nuclear and cytosolic ROI was generated from the DAPI and YAP-stained channels respectively, through segmentation by Otsu method thresholding⁷². An area outside the cell on the YAP-stained channel was selected as ROI for background signal. The area and sum intensity in the ROIs were measured and exported for analysis. Background was subtracted from the nuclear and cytosolic sum intensities, then the nuclear intensity was divided by the cytosolic intensity to obtain the nuclear:cytosolic ratio of YAP.”

c. Analysing only 30 cells for this experiment feels very low for this kind of experiment. The n number should be increased.

We have increased the N number to 100 cells for **Fig. 7c**. Consistently, we observed the ratio of nuclear:cytosol YAP significantly decreased in response to increasing substrate rigidity in zyxin-overexpressing D3 cells (Fig. 7c). In addition, we noticed slightly decreased nuclear

YAP in control cells (Fig. 7c-control), suggesting crosstalk with other signals in the regulation of YAP in rigidity sensing. We have updated **Fig. 7c** and **figure legend (Line 880 – Line 882)**:

“Fig. 7: c Nucleus/Cytosol YAP ratio in D3 cells transfected with either control or mCherry-Zyxin plasmids were analysed and plotted. Kruskal-Wallis test was used to test the differences among the three rigidities (N=100).”

d. Yap seems to move out from the nuclei in a stiff environment in zyxin expressing cells. This is surprising, and the authors should discuss this.

Dupont and his co-workers (Dupont et al., 2011) were the first group to report that cells spreading on stiff substrates promotes YAP nuclear localization which is independent of Hippo cascade but requires Rho GTPases and cytoskeleton tensions. We have acknowledged the significance of this work in linking rigidity sensing and YAP activity. The **Discussion section** has been substantially re-worked to compare the consistency and discrepancy between our findings with Dupont’s study carried out in human specialized cells and human mesenchymal stem cells, as well as other studies carried out in mouse and human embryonic stem cells (**Line 348 – Line 367**):

“We have shown that YAP’s distribution in the nucleus decreased when zyxin overexpressing mESCs were plated on stiff substrates. Our observations contrast with a study carried out in human specialized cells and human mesenchymal stem cells (MSC) by Dupont et al., which shows that cells plated on stiff substrates promotes YAP nuclear localization in a Hippo-independent pathway but through RhoA-GTPases-mediated cytoskeleton tensions⁶¹. We observed a slight reduction of nuclear YAP in control cells on stiffer substrates (Fig. 7c-control), suggesting possible crosstalk with other signaling pathways in the regulation of YAP in addition to focal adhesion and cytoskeleton tensions. Our observations that phospho-YAP (Fig. 7e) increased on stiffer substrates and with synergistic effects with zyxin overexpression further support a role for zyxin in the regulation of YAP through the Hippo-LATS cascade⁴⁸. Possible involvement of other kinases such as Akt and NDR1/2 has not escaped our attention^{59,60}. In line with our observations in mESCs, it has been reported that YAP localization in the nucleus significantly decreased in differentiated D3 cells³⁵ and E14 cells³⁶ which exhibit more well-spread morphology. Our findings suggest that zyxin overexpressing mESCs resemble early differentiated cells. Therefore, apart from mechanotransduction, early differentiation is another factor to be considered in our current experimental system. The discrepancy between our observations and those reported by Dupont et al., might be explained through cell type differences in multi-levels of cell potency. Moreover, mESCs differ from hESCs in self-renewal and pluripotency maintenance mechanisms⁶². Therefore, more studies are needed to determine if our findings could be applied to the human stem cell system.”

e. If Yap controls the Oct4 expression downstream of zyxin, it is surprising to see that Oct4 levels decline in the 6 kPa environment in zyxin expressing cells (Figure 7f.). The authors need to explain this or take this in consideration in their model (7g).

We thank reviewer for pointing it out. We have explained the observations in the **Results section (Line 289 – Line 295)** and revised our model (**Fig. 7g**):

“We observed overall Oct4 reduction in zyxin overexpressing D3 (Fig. 7f) and E14 (Fig. S6k) cells, which is consistent with our earlier findings showing that Oct4 is reduced upon zyxin overexpression (Fig. 4a-b), suggesting zyxin might play a role in regulating pluripotency through biochemical signaling. In addition, Oct4 level was further reduced in zyxin overexpressing cells on stiffer substrate (Fig. 7f, Fig. S6k), suggesting zyxin-YAP-mediated mechanical signaling as an additional and/or alternative regulation of Oct4 levels.”

We also compare our findings with another study carried out in *Xenopus laevis* embryos and HEK293 cells, which suggests that zyxin plays a significant role in regulating embryonic stem cell fate through biochemical signaling. See **Discussion section (Line 319 - Line 327)**:

*“An earlier study carried out in *Xenopus laevis* embryos and HEK293 cells also suggests that zyxin plays a significant role in regulating embryonic stem cell fate⁴⁹. However, the proposed mechanisms differ from ours. In this earlier study, it is reported that zyxin negatively regulates pluripotency genes through inhibiting the binding of Ybx1 mRNA-stabilizing protein to Oct4 mRNA, resulting in the degradation of Oct4 mRNA⁴⁹. The presence of nuclear zyxin may imply its participation in regulating gene expression^{50,51}. Nevertheless, we found that overexpressed zyxin is primarily localized to the FA instead of the nucleus (Fig. 5c-d), suggesting that zyxin-mediated mechano-transduction might be another mechanism in pluripotency regulation.”*

6. The authors present most of the data as bar graphs with SEM error bars. This does not give the complete picture of the data distribution. Instead, individual data points should be shown, and the variability of the data should be presented as a standard deviation. If SEM is used, it should be justified.

We have re-plotted all data to show individual data point. If the sample size is small, a bar chart is used with individual data point overlaid on the bars (**Fig. 2b, Fig. 3b, Fig. 4c-d, Fig. 7e-f, Fig. S2b, Fig. S6j-k**). Error bars represent the standard deviation. The remaining bar graphs were converted to box-and-whisker-plot with full data distribution (**Fig. 1d-g, Fig. 3e, Fig. 5e-h, Fig. 6, Fig. 7c, Fig. S4e-h, Fig. S5b-c, Fig. S6b-c/e-f/h, Fig. S7**). More statistical details were updated in **Method section (Line 567 – Line 578)** and **individual figure legend**.

“Statistics and Reproducibility

*Statistical analysis was performed with the GraphPad Prism version 9.4.1 for Windows, GraphPad Software, San Diego, California USA, www.graphpad.com. All data was subject to the Shapiro-Wilk normality test before the respective statistical methods were carried out. Statistical significance between two groups was determined through two-tailed unpaired Student’s t-test or Mann-Whitney U-test. Between three groups, it was determined through the one-way ANOVA or Kruskal-Wallis test. At least three independent experiments were analyzed. P-values less than or equal to 0.05 were considered significant. * $P \leq 0.05$; ** $P \leq 0.01$; *** $P \leq 0.001$; **** $P \leq 0.0001$; ns, not significant. Error bars represent standard deviations. In the box and whisker plots, the centre line is the median, the box-bounds are the 25th and 75th percentiles, and the whiskers are the 0.05 and 0.95 percentiles.”*

Minor comments:

1. The use of a red-green colour combination should be avoided when presenting the microscopy data.

We have changed all red-green colours to magenta-green colours.

2. The authors should state in the figures or in the legends the imaging plane and possible image projections, environment stiffnesses (Figure 6 c and d) and the statistical test used.

(a) We have updated the figure legends with “z-stacked images with maximum intensity projection were shown” in the following figure legends: **Fig. 1, Fig. 2, Fig. 3, Fig. 5, Fig. 7, Fig. S4, Fig. S6.**

(b) We have updated the figure legends with “projected cell area” and “projected focal adhesion area” in the following figure legends: **Fig. 1, Fig. 5, Fig. S4.**

(c) The stiffness used for **Fig. 6a-d** is 14 kPa. We have added the detail into **Fig. 6 legend (Line 857 – Line 858):**

“Fig. 6: a D3 and b E14 control cells and zyxin overexpression cells seeded on 14 kPa substrates were subjected to traction stress analysis.”

(d) Statistical test details were updated in **Method section (Line 567 – Line 578)** and the **individual figure legend**. See point number 6.

3. Authors should make sure that the methods section is up to date. For example, a detailed description of focal adhesion analysis, overexpression experiments and fluorescence intensity measurement is missing.

We have updated the **Methods section** as shown in **Line 391 – Line 578.**

4. Authors should adjust their use of the following words in the text and figures and aim for accurate description:

a. Size vs area. For example, when measuring the cell area from a single plane or projected images does not give the size of the cell.

We have changed “cell size” “focal adhesion size” to “cell area” “focal adhesion area” throughout the manuscript.

b. Cortex. When describing the localisation of actin fibres.

We have changed “cell cortex” to “cell periphery” throughout the manuscript.

c. Enhance. When describing the growth of focal adhesions or abundance of F-actin fibres.

We have made the necessary changes to be more specific.

Reviewers' comments:

Reviewer #1 (Remarks to the Author):

The authors have addressed the comments thoroughly. After addressing the minor comment below, I recommend the manuscript for publication.

1. Fig.S3: The control beta-tubulin level varies across samples, making it hard to draw conclusions from the western blots. Please perform quantification for the Oct4 and Zyxin blots, with normalization against beta-tubulin (similar to Fig.S2).

Reviewer #2 (Remarks to the Author):

First, I want to thank the authors for conducting the new experiments and revising the text. Overall, I think the new experimental data, revised text and updated model improve the manuscript's quality. However, considering the newly added data, I will still raise a few issues.

1. Authors have now repeated some of the key experiments on laminin, and they see a similar trend in enforcing the FAs upon zyxin overexpression. However, the cells on laminin seem more mechanosensitive than cells on FN (F6 and s7).

a. The authors say in the manuscript's revised version (rows 251-252) "Consistent with previous findings using fibronectin, zyxin overexpressing D3 cells seeded on laminin were more sensitive to increasing substrate rigidities". The statistical test is currently performed between the 6kPa vs 31kPa condition inside the CTRL or OE group. It would be beneficial to perform suited tests between the CTRL and OE in both LM and FN conditions.

b. Minor suggestion: Considering that CTRL cells on LM seem to a certain degree to sense the stiffness (s7 b and e), it would be beneficial to see what happens to Yap1 nuclear localisation (or phosphorylation) in the same conditions.

2. As requested, the authors have now increased the number of cells analysed for the nuc/cyt Yap quantification in Figure 7. Also, they mention (row 273-275) "In contrast to control cells, the ratio of nuclear:cytosol YAP significantly decreased in response to increasing substrate rigidity in zyxin-overexpressing D3 cells (Fig. 7c)". This statement is not fully supported by the data.

a. The control cells seem to also respond to the substrate stiffness (according to the statistics, this is significant 6kPa vs 31kPa). Therefore, there should be statistical testing comparing the CTRL and OE groups to support this claim.

3. The authors have completed the KD experiment that was requested in the first round of revisions. However, the analysis of this experiment has not been completed.

a. It would be beneficial to see the numbers and statistics from the Western blots in order to solidify the claim " While RA-treatment led to significant reduction of Oct4 in control cells, Oct4 levels were better maintained in RA-treated zyxin-knockdown cells (Fig. S3). These results suggest a role of zyxin in pluripotency maintenance."

b. Minor suggestion: Further, this would be a great set-up to investigate whether the KD cells are able to spread during RA treatment and if either the pYap/Yap ratio would be further declined or n/c yap ratio increase. However, I understand if the authors think these experiments are outside this manuscript's scope.

Reviewers' comments:

Reviewer #1 (Remarks to the Author):

The authors have addressed the comments thoroughly. After addressing the minor comment below, I recommend the manuscript for publication.

1.Fig.S3: The control beta-tubulin level varies across samples, making it hard to draw conclusions from the western blots. Please perform quantification for the Oct4 and Zyxin blots, with normalization against beta-tubulin (similar to Fig.S2).

We thank the reviewer for the suggestion. We have performed quantification (normalized against β -tubulin) for Zyxin (new Fig. S3d) and Oct4 (new Fig. S3e). Our results suggest that zyxin knockdown could better maintain Oct4 levels at 72H (Fig. S3e-72H). The statistical results were presented in **Fig. S3** and more statistical details were updated in **Fig. S3 figure legend. Results section (Line 171 - Line 173)** was also updated, see below.

“At 72H time point, while RA-treatment led to significant reduction of Oct4 in control cells, Oct4 levels were better maintained in RA-treated zyxin-knockdown cells (Fig. S3e-72H). These observations suggest possible roles for zyxin in pluripotency maintenance.”

Reviewer #2 (Remarks to the Author):

First, I want to thank the authors for conducting the new experiments and revising the text. Overall, I think the new experimental data, revised text and updated model improve the manuscript's quality. However, considering the newly added data, I will still raise a few issues.

1. Authors have now repeated some of the key experiments on laminin, and they see a similar trend in enforcing the FAs upon zyxin overexpression. However, the cells on laminin seem more mechanosensitive than cells on FN (F6 and s7).
a. The authors say in the manuscript's revised version (rows 251-252) "Consistent with previous findings using fibronectin, zyxin overexpressing D3cells seeded on laminin were more sensitive to increasing substrate rigidities". The statistical test is currently performed between the 6kPa vs 31kPa condition inside the CTRL or OE group. It would be beneficial to perform suited tests between the CTRL and OE in both LM and FN conditions.

We thank the reviewer for the suggestions. In addition to one-way ANOVA test (or Kruskal-Wallis test) to analyse the differences among three rigidities, we have performed additional two-tailed unpaired Student's *t*-test (or Mann-Whitney *U*-test) to compare the differences between Ctrl and Zyx O/E cells seeding on individual rigidity. The additional statistical analysis was applied to both Fibronectin (**Fig. 6f and h, Fig. S6b and d**) and laminin (**Fig. S7b and d**) conditions. The statistical details were updated in the **respective figure legends**.

b. Minor suggestion: Considering that CTRL cells on LM seem to a certain degree to sense the stiffness (s7 b and e), it would be beneficial to see what happens to Yap1 nuclear localisation (or phosphorylation) in the same conditions.

Dupont's study (Dupont et al., 2011) suggests that YAP localization is mainly determined by cell spreading imposed by the matrix. Since D3 cells exhibited similar spreading, actin and focal adhesion morphologies when plated on fibronectin (Fig. 5) and laminin (Fig. S4), we predicted that YAP localization would likely be similar in both fibronectin and laminin conditions. Nevertheless, whether changes on ECM types that impact traction stress magnitudes/directions (Fig. 6f,h and Fig. S7b,d) will contribute to differences in YAP localization/phosphorylation remain to be investigated. Apart from fibronectin and laminin, it would be also interesting to compare mechanosensing, cell spreading, YAP localization and changes on pluripotency when mESCs are seeding on different ECMs, e.g. collagen and vitronectin. These comparison could be insightful to establish the roles of ECM in regulating mESC mechanosensing and pluripotency. However, it is out of the scope of this current study which focuses more on zyxin. Nevertheless, we have proposed looking into different ECMs and their roles in mechanosensing and pluripotency regulation as future work in the **Discussion section (Line 348 – Line 352)**:

“Apart from fibronectin and laminin, more studies can be conducted to compare mechanosensing, cell spreading, YAP localization and changes on pluripotency when mESCs are seeded on different ECMs, e.g. collagen and vitronectin. These comparisons are likely to provide more insights on the roles of ECM in regulating mESC mechanosensing and pluripotency.”

2. As requested, the authors have now increased the number of cells analysed for the nuc/cyt Yap quantification in Figure 7. Also, they mention (row 273-275) "In contrast to control cells, the ratio of nuclear:cytosol YAP significantly decreased in response to increasing substrate rigidity in zyxin-overexpressing D3 cells (Fig. 7c)". This statement is not fully supported by the data.

a. The control cells seem to also respond to the substrate stiffness (according to the statistics, this is significant 6kPa vs 31kPa). Therefore, there should be statistical testing comparing the CTRL and OE groups to support this claim.

We thank the reviewer for the suggestion. We have performed additional Mann-Whitney *U*-test to compare the differences between Ctrl and Zyx O/E cells seeded on individual rigidity (Fig. 7c). More statistical testing supports our observations that the ratio of nuclear:cytosol YAP significantly decreased in response to increasing substrate rigidity in zyxin-overexpressing cells compared to control cells. The statistical details and tests conducted were updated in the **Fig. 7c figure legend**.

3. The authors have completed the KD experiment that was requested in the first round of revisions. However, the analysis of this experiment has not been completed.

a. It would be beneficial to see the numbers and statistics from the Western blots in order to solidify the claim " While RA-treatment led to significant reduction of Oct4 in control cells, Oct4 levels were better maintained in RA-treated zyxin-knockdown cells (Fig. S3). These results suggest a role of zyxin in pluripotency maintenance.”

We thank the reviewer for the suggestion. We have performed quantification and statistical analysis for Zyxin (new Fig. S3d) and Oct4 (new Fig. S3e). Our results suggest that zyxin knockdown could better maintain Oct4 levels at 72H (Fig. S3e-72H). The statistical results were presented in **Fig. S3** and more statistical details were updated in **Fig. S3 figure legend. Results section (Line 171 - Line 173)** was also updated:

“At 72H time point, while RA-treatment led to significant reduction of Oct4 in control cells, Oct4 levels were better maintained in RA-treated zyxin-knockdown cells (Fig. S3e-72H). These observations suggest possible roles for zyxin in pluripotency maintenance.”

b. Minor suggestion: Further, this would be a great set-up to investigate whether the KD cells are able to spread during RA treatment and if either the pYap/Yap ratio would be further declined or n/c yap ratio increase. However, I understand if the authors think these experiments are outside this manuscript's scope.

In our previous study carried out in fibroblasts (Yip et al., 2021), we showed that zyxin knockdown cells were not able to spread as well as control cells on the same substrate. Till date, very little work has been performed using mESC as a model system. We agree with the reviewer that it would be interesting to expand our study in the future using our current set-up with the KD cells. The suggested experiments are proposed as future work in the **Discussion section (Line 346 – Line 348)**:

“In addition to zyxin overexpression experiments, it would be interesting to investigate zyxin knockdown effects on cell spreading and YAP localization/phosphorylation in mESCs.”